# A Tensor Space for Multi-View and Multitask Learning Based on Einstein and Hadamard Products: A Case Study on Vehicle Traffic Surveillance Systems

**DOI:** 10.3390/s24237463

**Published:** 2024-11-22

**Authors:** Fernando Hermosillo-Reynoso, Deni Torres-Roman

**Affiliations:** Center for Research and Advanced Studies of the National Polytechnic Institute, Department of Electrical Engineering and Computer Sciences, Telecommunications Section, Av. del Bosque 1145, El Bajio, Zapopan 45019, Jalisco, Mexico; deni.torres@cinvestav.mx

**Keywords:** Einstein product, Hadamard product, Hadamard factor tensors, multi-view learning, multitask learning, vehicle traffic surveillance

## Abstract

Since multi-view learning leverages complementary information from multiple feature sets to improve model performance, a tensor-based data fusion layer for neural networks, called Multi-View Data Tensor Fusion (MV-DTF), is used. It fuses M feature spaces $X_{1},\cdots ,X_{M}$, referred to as views, in a new latent tensor space, S, of order *P* and dimension $J_{1}\times \cdots \times J_{P}$, defined in the space of affine mappings composed of a multilinear map $T:X_{1}\times \cdots \times X_{M}\to S$—represented as the Einstein product between a (P+M)-order tensor A anda rank-one tensor, $X=x^{\left(1\right)}⊗\cdots ⊗x^{\left(M\right)}$, where $x^{\left(m\right)}\in X_{m}$ is the *m*-th view—and a translation. Unfortunately, as the number of views increases, the number of parameters that determine the MV-DTF layer grows exponentially, and consequently, so does its computational complexity. To address this issue, we enforce low-rank constraints on certain subtensors of tensor A using canonical polyadic decomposition, from which *M* other tensors $U^{\left(1\right)},\cdots ,U^{\left(M\right)}$, called here Hadamard factor tensors, are obtained. We found that the Einstein product $A{⊛}_{M}X$ can be approximated using a sum of *R* Hadamard products of *M* Einstein products encoded as $U^{\left(m\right)}{⊛}_{1}x^{\left(m\right)}$, where *R* is related to the decomposition rank of subtensors of A. For this relationship, the lower the rank values, the more computationally efficient the approximation. To the best of our knowledge, this relationship has not previously been reported in the literature. As a case study, we present a multitask model of vehicle traffic surveillance for occlusion detection and vehicle-size classification tasks, with a low-rank MV-DTF layer, achieving up to 92.81% and 95.10% in the normalized weighted Matthews correlation coefficient metric in individual tasks, representing a significant 6% and 7% improvement compared to the single-task single-view models.

## 1. Introduction

Vehicle traffic surveillance (VTS) systems are key components of intelligent transportation systems (ITSs), as they enable the automated video content analysis of traffic scenes to extract valuable traffic data. It includes crucial aspects of vehicle behavior, such as trajectories and speed, as well as traffic parameters, e.g., lane occupancy, traffic volume, and density. These data serve as the cornerstone for a variety of high-level ITS applications, including collision detection [1,2], route planning, and traffic control [3,4]. Currently, there exist several mathematical models for various tasks related to vehicle traffic, each with different conditions and traffic network topologies. For a comprehensive overview of vehicle traffic models, see, e.g., [5].

However, due to the complex nature of vehicle traffic, VTS systems are usually broken down into a set of smaller tasks, including vehicle detection, occlusion handling, and classification [6,7,8,9,10,11,12,13,14]. Each task is represented as a feature model, which should be related to the underlying task-specific explanatory factors, while it is either developed by human experts (hand-crafted) or automatically learned. These features focus on specific aspects of vehicles, such as texture, color, and shape, which individually provide complementary information to each other. Therefore, finding a highly descriptive feature model is crucial for enhancing the learning process on every VTS task.

Such feature diversity has made data fusion (DF) attractive for leveraging its shared and complementary information. DF allows for the integration of data from different sources to enhance our understanding and analysis of the underlying process [15]. In this context, there are two common DF levels [16]: low-level, where data are combined before analysis, and decision-level, where processed data from each source are integrated at a higher level, such as in ensemble learning [17]. Moreover, the diverse nature of data sources poses challenges, such as heterogeneity across sources, high-dimensional data, missing values, and a lot of redundancy that DF algorithms should address [18,19].

As part of DF, multi-view learning (MVL) is a machine learning (ML) paradigm that exploits the shared and complementary information contained in multiple data sources, called views, obtained from different feature sets [20]. Here, data represented by *M* views are referred to as *M*-view data. For instance, an image represented by texture, edges, and color features can be regarded as three-view data. MVL methods can be grouped into three categories: co-training, multiple kernel learning, and subspace learning (SL) [21,22]. Among these, SL-based methods focus on learning a low-dimensional latent subspace that captures the shared information across views [23].

On the other hand, multitask learning (MTL) is another ML paradigm where multiple related tasks are learned simultaneously to leverage their shared knowledge, with the ultimate aim of improving generalization and performance in individual tasks [24,25,26,27,28].

Recently, artificial neural networks (ANNs) have shown superior performance in vision-based VTS systems. ANNs are computational models built from a composition of functions, called layers, which together capture the underlying relationships between the so-called input and output spaces to solve a given task, such as regression or classification [29]. Such layers, including fully connected (FC) and convolutional (Conv), are parameterized by weights and biases structured as tensors, matrices, or vectors, which are learned during training. Notably, the first layers usually act as feature extractors, whereas higher layers capture the relationships between extracted features and the output space.

Furthermore, higher-order tensors [30], or multidimensional arrays, have gained significant attention over the last decade due to their ability to naturally represent multi-modal data, e.g., images and videos, and their interactions. They have been successfully applied in various domains, including signal processing [31], machine learning [32,33,34,35,36,37], computer vision [38], and wireless communications [39,40]. For instance, tensor methods such as decomposition models have been employed for the low-rank approximation of tensor data, enabling more efficient and effective analysis of such data.

In this work, we propose a computationally efficient tensor-based multi-view data fusion layer for neural networks, here expressed as the Einstein product. Our approach leverages multiple feature spaces to address the limitations inherent to single-view models, such as reduced data representation capacity and model overfitting. It offers improved flexibility and scalability, as it enables the integration of additional views without significantly increasing the computational burden. Finally, we present a case study with a multitask, multi-view VTS model, demonstrating significant performance improvements in vehicle-size classification and occlusion-detection tasks.

### 1.1. Related Work

Occlusion detection is a challenging problem in vision-based tasks, in which vehicles or some parts of them are hidden by other elements in the traffic scene, making their detection a difficult task. Early works have explored approaches based on empirical models, which infer the presence of occlusion by assuming specific geometric patterns, such as concavity in the shape of occluded vehicles [41,42,43,44,45,46,47]. Recently, deep learning (DL) has also been employed for occlusion detection [48,49,50,51,52], where such models are even capable of reconstructing the occluded parts [53,54].

Several algorithms based on ML and DL have been proposed for intra- and inter-class vehicle classification [6,8,9,55,56,57,58,59,60]. In [8], Hsieh et al. employ the optimal classifier to categorize vehicles as cars, buses, or trucks by leveraging the linearity and size features of vehicles, achieving accuracy of up to 97.0%. Moussa [9] introduces two levels of vehicle classification: the multiclass level, which categorizes vehicles as small, midsize, and large, and the intra-class level, in which midsize vehicles are classified as pickups, SUVs, and vans. In [6], we proposed a one-class support vector machine (OC-SVM) classifier with a radial basis kernel to classify vehicles as small, midsize, and large. By representing vehicles in a 3D feature space (area, width, and aspect-ratio) features, a recall, precision, and f-measure of up to 99.05% were achieved for the midsize class. Other techniques include the gray-level co-occurrence matrix (GLCM) [61], 3D appearance models [62,63,64], eigenvehicles [65], and non-negative factorization [66,67,68]. Recently, CNN-based classifiers have been employed, outperforming previous works [55,58,59,60,69].

Other works based on MLV and MTL have also been developed for VTS systems. For instance, Wang et al. [70] proposed an MVL approach to foreground detection, where three-view heterogeneous data (brightness, chromaticity, and texture variations) are employed to improve detection performance. Then, their conditional probability densities are estimated via kernel density estimation, followed by pixel labeling through a Markov random field. In [71], a multi-view object retrieval approach to surveillance videos integrates semantic structure information from CNNs trained on ImageNet and deep color features, using locality-sensitive hashing (LSH) to encode the features into short binary codes for efficient retrieval. Chu et al. [72] present vehicle detection with multitask CNNs and a region-of-interest (RoI) voting scheme. This framework addresses simultaneously supervision with subcategory, region overlap, bounding-box regression, and category information to enhance detection performance. In [73], a multi-task CNN for traffic scene understanding is proposed. The CNN consists of a shared encoder and specific decoders for road segmentation and object detection, generating complementary representations efficiently. Additionally, the detection stage predicts object orientation, aiding in 3D bounding box estimation. Finally, Liu et al. [74] introduce the Multi-Task Attention Network (MTAN), a shared network with a global feature pooling and task-specific soft-attention modules to learn task-specific features from global features while allowing feature sharing across tasks.

Although, our work is focused on multi-view and multitask VTS systems, some works related to other domains are also overviewed. In [36], a tensor-based, multi-view feature selection method called DUAL-TMFS is proposed for effective disease diagnosis. This approach integrates clinical, imaging, immunologic, serologic, and cognitive data into a joint space using tensor products, and it employs SVM with recursive feature elimination to select relevant features, improving classification performance in neurological disorder datasets. Zadeh et al. [75] introduce a novel model called a tensor fusion network for multimodal sentiment analysis. It leverages the outer product between modalities to model both the intra-modality and inter-modality dynamics. On the other hand, Liu et al. [76] propose an efficient multimodal fusion scheme using low-rank tensors. Experimental validations across multimodal sentiment analysis, speaker trait analysis, and emotion recognition tasks demonstrate competitive performance and robustness across a variety of low-rank settings.

Table 1 offers a comprehensive overview of existing research related to our approach and to VTS systems. It highlights the use of ML and DL approaches, fed either by hand-crafted features or raw data with automatic feature learning, to capture the underlying task patterns. While DL features generally achieve superior performance, they require large, high-quality training sets and high computational complexity models to find suitable representations. Conversely, hand-crafted features can perform competitively for specific tasks, but determining the optimal feature representation is challenging, as no single hand-crafted feature can fully describe the underlying task’s relationships.

Furthermore, the emerging trend towards the adoption of ANN models on VTS systems is evident. However, despite their high performance, these models demand substantial memory and computational resources for learning and inference, as their layers are usually overparameterized. To address these challenges, various techniques such as sparsification, quantization, and low-rank approximation have been proposed to compress the parameters of pre-trained layers [77,78,79,80,81,82,83]. Among these techniques, low-rank approximation is very often employed. In [79,80], Denil et al. compress FC layers using matrix decomposition models. Conv layers are compressed via tensor decompositions, including canonical polyadic decomposition (CPD) [81,82] and Tucker decomposition [83]. However, compressing pre-trained layers usually results in an accuracy loss, and a fine-tuning procedure is often employed to recover the accuracy drop [82,84,85,86]. Therefore, some authors have suggested the incorporation of low-rank constraints into the optimization problem [87,88,89]. Other works have found that compressing raw images before training also contributes to computational complexity reduction, as suggested in [32,90]. Additionally, in [91], tensor contraction layers (TCLs) and tensor regression layers (TRLs) are introduced in CNNs for dimensionality reduction and multilinear regression tasks, respectively. This approach imposes low-rank constraints via Tucker decomposition on the weights of TCLs and TRLs to speed up their computations.

**Table 1 sensors-24-07463-t001:** Related work summary.

Reference	Input	Method	Contribution
[6,8,9,10,14]	Single-view	ML	Hand-crafted geometric features represent vehicles for detection and classification using ML-based algorithms
[11,12]	Single-view	DL	CNN models are proposed to perform automatic feature learning for vehicle detection and classification
[65]	Single-view	Eigenvalue decomposition	Eigenvehicles are introduced as an unsupervised feature representation method for vehicle recognition
[66,67,68]	Single-view	Nonnegative factorization	A part-based model is employed for vehicle recognition via non-negative matrix/tensor factorization
[72,73,74]	Single-view	DL-based MTL	MTL models based on DL are employed to simultaneously perform multiple tasks, including road segmentation, vehicle detection and classification
[92]	Multi-view	DL	This work employs a YOLO-based model that fuses camera and LiDAR data at multiple levels
[61,93,94]	Single-view	ML	Single-view features, such as HOG, Haar wavelets, or GLCM, represent vehicles for classification in ML models
[95]	Multi-view	Tucker decomposition	A tensor decomposition is employed for feature selection of HOG, LBP, and FDF features
[70,71,96]	Multi-view	MVL	MVL approaches are proposed to enhance vehicle detection, classification, and background modeling by learning richer data representations from color features
[30,97,98,99,100]	−	−	These works provide theoretical foundations on tensors and its operations, such as the Einstein and Hadamard products, with applications across multiple domains
[32,77,78,79,80,81,82,83,90]	−	DL	Matrix and tensor decompositions are employed for speeding up CNNs by compressing FC and Conv layers and reducing the dimensionality of their input space
[91]	−	DL	Multilinear layers are introduced for dimensionality reduction and regression purposes in CNNs, leveraging tensor decompositions for efficient computation.

### 1.2. Contributions

The main contributions of this work are the following:We found a novel connection or mathematical relationship between the Einstein and Hadamard products for tensors (for details, see Section 5.2). From this connection, other algorithms for efficient approximations of the Einstein product can be developed.Since multi-view models provide a more comprehensive input space than single-view models, we employ a tensor-based data fusion layer, here called multi-view data tensor fusion (MV-DTF). Unlike other works, our approach maps the multiple feature spaces (views) into a latent tensor space, S, using a multilinear map, here expressed as the Einstein product (see Section 5), followed by a translation.A major drawback of the MV-DTF layer is its high computational complexity, which grows exponentially with the number of views. To address this issue, a low-rank approximation for the MV-DTF layer, here called the low-rank multi-view data tensor fusion (LRMV-DTF) layer, is also proposed. This approach leverages the novel relationship between the Einstein and Hadamard products (see Section 5.2), where the lower the rank values, the more computationally efficient the operation.As a case study, we introduce a high-performance multitask ANN model for VTS systems capable of simultaneously addressing various VTS tasks but which is here limited to occlusion detection and vehicle-size classification. This model incorporates the proposed LRMV-DTF layer as multi-view feature extractor to provide a more comprehensive input space compared to individual spaces.

### 1.3. How to Read This Article

For a comprehensive understanding of this paper, the following is suggested the following: Section 1 presents the motivation behind our research on VTS systems, as well as a review of their related works, while Section 2 introduces tensor algebra and multilinear maps, which will be essential for understanding the subsequent mathematical definitions; however, if you are already familiar with their theoretical foundations, you can proceed directly to Section 3 to delve into the problem statement and its mathematical formulation, where the main objectives are stated. These objectives are important to understand the major results of the paper. Section 4 provides a comprehensive overview of VTS systems and their associated tasks as an important case study. If you are already familiar with these concepts, proceed to Section 5 for the technical and mathematical details of the MV-DTF layer. Particularly, Section 5 is very important because it presents the novel connection between Einstein and Hadamard products. Section 6 presents the results and their analysis for a deeper understanding of our findings, which are complemented by figures and tables to facilitate data interpretation. Finally, Section 7 provides the conclusions of this work, summarizing the key points and suggesting directions for future research.

## 2. Mathematical Background

### 2.1. Notation

In this study, we adopt the conventional notation established in [30], along with other commonly used symbols. Table 2 provides a comprehensive overview of the symbols utilized in this paper. An *N*th-order tensor is denoted by $X\in R^{I_{1}\times \cdots \times I_{N}}$, where the dimension $I_{n}$ is usually referred to as the *n*-mode of X. The *i*th entry of a vector, $x\in R^{I}$, is denoted as $x_{i}$; the (i,j)th entry of a matrix $X\in R^{I\times J}$ by $x_{\mathrm{ij}}$; while the $\left(i_{1},\cdots ,i_{N}\right)$th entry of an *N*th-order tensor $X\in R^{I_{1}\times \cdots \times I_{N}}$ is denoted as $x_{i_{1}i_{2}\ldots i_{N}}$, where $i_{n}\in \left[I_{n}\right]$ is called the *n*-mode index. The *n*-mode fiber of an *N*th-order tensor is an $I_{n}$-dimensional vector resulted from fixing every index but $i_{n}$; i.e., $X_{i_{1}i_{2}\ldots i_{n-1}:i_{n+1}\ldots i_{N}}\in R^{I_{n}}$, where colon mark: denotes all possible values of the *n*-mode index $i_{n}$, i.e., $\left[I_{n}\right]$. The *i*th *n*-mode slice of an *N*th-order tensor is an (N−1)th-order tensor defined by just fixing the $i_{n}$ index, i.e., $X_{::\cdots i_{n}:\cdots :}$. Finally, for any two functions, $f_{1}$ and $f_{2}$, $f_{1}\circ f_{2}$ denotes their function composition. For an understanding on tensor algebra, we refer the interested reader to the comprehensive work by Kolda and Bader [30].

### 2.2. Multilinear Algebra

This section provides an overview of basic concepts of multilinear algebra, such as tensors and their operations over a set of vector spaces.

**Definition** **1**(Multilinear map [101])**.** *Let $V_{1},\cdots ,V_{M}$ and W be vector spaces over a field, R. And let $T:V_{1}\times \cdots \times V_{M}\to W$ be a function that maps an ordered M-tuple of vectors, $\left(v^{\left(1\right)},\cdots ,v^{\left(M\right)}\right)\in V_{1}\times \cdots \times V_{M}$, into an element, w∈W, where $v^{\left(m\right)}\in V_{m}\forall m\in \left[M\right]$. If, for all a,b∈R and $v^{\left(m\right)},u^{\left(m\right)}\in V_{m}\forall m\in \left[M\right]$, Equation (Equation 1) holds, then T is said to be a multilinear map (or an M-linear map); i.e., it is linear in each argument.*
(1)$$\begin{array}{c} T(v^{\left(1\right)},\cdots ,v^{\left(m-1\right)},a\;v^{\left(m\right)}+bu^{\left(m\right)},v^{\left(m+1\right)},\cdots ,v^{\left(M\right)})=\\ aT\left(v^{\left(1\right)},\cdots ,v^{\left(m-1\right)},v^{\left(m\right)},v^{\left(m+1\right)}\cdots ,v^{\left(M\right)}\right)+bT\left(v^{\left(1\right)},\cdots ,v^{\left(m-1\right)},u^{\left(m\right)},v^{\left(m+1\right)}\cdots ,v^{\left(M\right)}\right) \end{array}$$

**Definition** **2**(Tensor product)**.** *Let $V_{1},\cdots ,V_{M}$ and W be real vector spaces, where $\dim\left(V_{m}\right)=I_{m}\forall m\in \left[M\right]$, and dim(W)=J. Then, the tensor product of the set of M vector spaces $V_{1},\cdots ,V_{M}$, denoted as $V_{1}⊗\cdots ⊗V_{M}$, is another vector space of dimension $\dim\left(V_{1}⊗\cdots ⊗V_{M}\right)=\prod_{m=1}^{M}\dim\left(V_{m}\right)$, called tensor space, together with a multilinear map, $\pi :V_{1}\times \cdots \times V_{M}\to V_{1}⊗\cdots ⊗V_{M}$, that satisfies the following universal mapping property* [101,102]*: for any multilinear map $T:V_{1}\times \cdots \times V_{M}\to W$, there exists a unique linear map, $\Phi :V_{1}⊗\cdots ⊗V_{M}\to W$, such that T=Φ∘π.*

**Definition** **3**(Tensor)**.** *Let $V_{1},\cdots ,V_{M}$ be vector spaces over some field, F, where $\dim\left(V_{m}\right)=I_{m}\forall m\in \left[M\right]$. An M-order tensor, denoted as X, is an element in the tensor product $V_{1}⊗\cdots ⊗V_{M}$.*

**Definition** **4**(*m*-mode matricization [30])**.** *The m-mode matricization is a mapping that rearranges the m-mode fibers of a tensor, $X\in R^{I_{1}\times \cdots \times I_{M}}$, into the columns of a matrix, $X_{\left(m\right)}\in R^{I_{m}\times J}$, where $J=\prod_{k=1,k\neq m}^{M}I_{k}$.*

**Definition** **5**(Rank-one tensor)**.** *Let $X\in R^{I_{1}\times \cdots \times I_{M}}$ be an Mth-order tensor, and let $x^{\left(1\right)},\cdots ,x^{\left(M\right)}$ be a set of M vectors, where $x^{\left(m\right)}\in R^{I_{m}}$ for all m∈[M]. Then, if X can be written using the tensor product $x^{\left(1\right)}⊗\cdots ⊗x^{\left(M\right)}$, it is said to be a rank-one tensor, and its $\left(i_{1},\cdots ,i_{M}\right)$-th entry will be determined by $x_{\subset }i_{1}\cdots i_{M}=\prod_{m=1}^{M}x_{i_{m}}^{\left(m\right)}$.*

**Definition** **6**(Tensor decomposition rank)**.** *The decomposition rank, R, of a tensor, $X\in R^{I_{1}\times \cdots \times I_{M}}$, is the smallest number of rank-one tensors that reconstructs X exactly as their sum. Then, X is called a rank-R tensor.*

**Definition** **7**(Tensor multilinear rank)**.** *For any Mth-order tensor, X, its multilinear rank, denoted as mlrank(X) is the M-tuple $\left(r_{1},\cdots ,r_{M}\right)$, whose mth entry, $r_{m}$, corresponds to the dimension of the column space of $X_{\left(m\right)}$, i.e., $r_{m}=\dim\left(\mathrm{Col}\left(X_{\left(m\right)}\right)\right)$, formally called m-mode rank.*

**Definition** **8**(Tensor *m*-mode product)**.** *Given a tensor, $X\in R^{I_{1}\times \cdots \times I_{M}}$, and a matrix, $U\in R^{J\times I_{m}}$, their m-mode product, denoted as $X\times_{m}U$, produces another tensor, $Y\in R^{I_{1}\times \cdots \times I_{m-1}\times J\times I_{m+1}\times \cdots \times I_{M}}$, whose $\left(i_{1},\cdots ,i_{m-1},j,i_{m+1},\cdots ,i_{M}\right)$th entry is given by Equation (Equation 2). Therefore, $Y=X\times_{m}U⟺Y_{\left(m\right)}=UX_{\left(m\right)}$.*
(2)$$y_{i_{1},\cdots i_{m-1}ji_{m+1}\cdots i_{M}}=\sum_{i_{m}=1}^{I_{m}}x_{i_{1},\cdots i_{m}\cdots i_{M}}\cdot u_{ji_{m}}$$

### 2.3. Einstein and Hadamard Products

In this section, the fundamental concepts for the mathematical modeling of the MV-DTF layer are presented, including the Hadamard and Einstein products.

**Definition** **9**(Inner product)**.** *For any two tensors, A, $B\in R^{I_{1}\times \cdots \times I_{M}}$, their inner product is defined as the sum of the product of each entry, as Equation (Equation 3) shows:*
(3)$$\langle A,B\rangle =\sum_{i_{1}=1}^{I_{1}}\sum_{i_{2}=1}^{I_{2}}\cdots \sum_{i_{M}=1}^{I_{M}}a_{i_{1}i_{2}\cdots i_{M}}b_{i_{1}i_{2}\cdots i_{M}}$$

**Definition** **10**(Hadamard product)**.** *The Hadamard product of two Nth-order tensors A, $B\in R^{I_{1}\times \cdots \times I_{M}}$, denoted as A⊙B, results in an Mth-order tensor, $C\in R^{I_{1}\times \cdots \times I_{M}}$, such that its $\left(i_{1},\cdots ,i_{M}\right)$th-entry $c_{i_{1}\cdots i_{M}}$ is equal to the element-wise product $a_{i_{1}\cdots i_{M}}\cdot b_{i_{1}\cdots i_{M}}$.*

**Definition** **11**(Einstein product [100,103])**.** *Given two tensors, $A\in R^{I_{1}\times \cdots \times I_{M}\times K_{1}\times \cdots \times K_{N}}$ and $B\in R^{K_{1}\times \cdots \times K_{N}\times J_{1}\times \cdots \times J_{P}}$, of order M+N and N+P, their Einstein product or tensor contraction, denoted as $A{⊛}_{N}B$, produces an M+P tensor, $C\in R^{I_{1}\times \cdots \times I_{M}\times J_{1}\times \cdots \times J_{P}}$, whose $\left(i_{1},\cdots ,i_{M},j_{1},\cdots ,j_{P}\right)$th entry is given by the inner product between subtensors $A_{i_{1}\cdots i_{M}:\cdots :}$ and $B_{:\cdots :j_{1}\cdots j_{P}}$, as Equation (Equation 4) shows:*
(4)$$c_{i_{1},\cdots ,i_{M},j_{1},\cdots ,j_{P}}=\sum_{k_{1}=1}^{K_{1}}\cdots \sum_{k_{l}=1}^{K_{N}}a_{i_{1}\cdots i_{M}k_{1}\cdots k_{N}}b_{k_{1}\cdots k_{N}j_{1}\cdots j_{P}}=\langle A_{i_{1}\cdots i_{M}:\cdots :},B_{:\cdots :j_{1}\cdots j_{P}}\rangle \;$$

The product $A{⊛}_{M}B$ can be understood as a linear map, $T:R^{I_{1}\times \cdots \times I_{M}}\to R^{J_{1}\times \cdots \times J_{P}}$; i.e., for any two scalars α,β∈R, and tensors $B_{1},B_{2}\in R^{K_{1}\times \cdots \times K_{N}\times J_{1}\times \cdots \times J_{P}}$, the following properties hold:Distributive: $A{⊛}_{N}\left(B_{1}+B_{2}\right)=A{⊛}_{N}B_{1}+A{⊛}_{N}B_{2}$.Homogeneity: $\alpha A{⊛}_{N}B_{1}=A{⊛}_{N}\alpha B_{1}=\alpha \left(A{⊛}_{N}B_{1}\right)$.

### 2.4. Subspace Learning

Recent advances in sensing and storage technologies have resulted in the generation of massive amounts of complex data, commonly referred to as big data [104,105]. These data are often represented in a high-dimensional space, making their visualization and analysis a challenging task. To address these challenges, subspace learning methods have emerged as a powerful approach to learning a low-dimensional representation of high-dimensional data [106,107], such as the spatial and temporal information encoded in videos. In this section, a brief review of linear and multilinear methods for subspace learning is presented, highlighting their advantages and disadvantages.

#### 2.4.1. Linear Subspace Learning (LSL)

Given a dataset, $\left\{x^{\left(1\right)},\ldots ,x^{\left(N\right)}\right\}$, of *N* samples, arranged in matrix form as $X\in R^{I\times N}$, whose *n*-th column vector corresponds to the *n*-th sample $x^{\left(n\right)}\in R^{I}$, LSL seeks to find a linear subspace of $R^{I}$ that best explains the data. The resulting subspace can be spanned by a set of J<I linearly independent basis vectors, $u_{1},\cdots ,u_{J}$, where $u_{j}\in R^{I}$. By leveraging this subspace, high-dimensional data can be projected onto a lower-dimensional space $R^{J}$, as Equation (Equation 5) shows:(5)$$G=U^{T}X=X\times_{1}U^{T}$$
where $U=\left[u_{1},\cdots ,u_{J}\right]\in R^{I\times J}$ is called the factor matrix, whose columns correspond to the basis vectors, and $G\in R^{J\times N}$ is the projection of the input matrix X onto U.

A wide variety of techniques have been proposed to address the LSL problem, ranging from unsupervised approaches such as principal component analysis [108], factor analysis (FA) [109], independent component analysis [110], canonical correlation analysis [111], and singular value decomposition [112], as well as supervised approaches like linear discriminant analysis [113]. Subsequently, such techniques aim to estimate U by solving optimization problems such as maximizing the variance or minimizing the reconstruction error of the projected data.

Although LSL methods have shown great effectiveness in modeling vector-based observations, they face difficulties when addressing multidimensional data. Then, to apply LSL methods on tensor data, it is necessary to vectorize them. Unfortunately, this transformation very often leads to a computationally intractable problem due to the large number of parameters to be estimated, and the model may suffer from overfitting. Furthermore, vectorization also destroys the inherent multidimensional structure and correlations across modes of tensor data [30,106].

#### 2.4.2. Multilinear Subspace Learning (MSL)

Multilinear subspace learning is a mathematical framework for exploring, analyzing, and modeling complex relationships over tensor data, preserving their inherent multidimensional structure. According to Lu [106], the MSL problem can be formulated as follows: Given a dataset $\left\{X^{\left(1\right)},\cdots ,X^{\left(N\right)}\right\}$ arranged in tensor form as $X\in R^{I_{1}\times \cdots \times I_{M}\times N}$, where subtensor $X_{:\cdots :n}$ corresponds with the *n*-th data point $X^{\left(n\right)}\in R^{I_{1}\times \cdots \times I_{M}}$, MLS seeks to find a set of *M* subspaces that best explains data, where the *m*th subspace resides in $R^{I_{m}}$ and is spanned by a set of $J_{m}<I_{m}$ linearly independent basis vectors, $u_{1}^{\left(m\right)},\cdots ,u_{J_{m}}^{\left(m\right)}\in R^{I_{m}}$. The MSL problem can be formally defined using Equation (Equation 6):(6)$$\underset{U^{\left(1\right)},\cdots ,U^{\left(M\right)}}{arg max}\Phi \left(U^{\left(1\right)},\cdots ,U^{\left(M\right)},X\right)$$
where $U^{\left(m\right)}=\left[u_{1}^{\left(m\right)},\ldots ,u_{J_{m}}^{\left(m\right)}\right]\in R^{I_{m}\times J_{m}}$ is a matrix whose columns correspond to the basis vectors of the *m*-th subspace, and Φ denotes a function to be maximized.

A classical MSL technique is the Tucker decomposition [30], which aims to approximate a given *M*th-order tensor, $X\in R^{I_{1}\times \cdots \times I_{M}}$, into a core tensor, $G\in R^{R_{1}\times \cdots \times R_{M}}$, multiplied along the *m*-mode by a matrix, $U^{\left(m\right)}$, for all m∈[M], as Equation (Equation 7) shows:(7)$$X≊G\times_{1}U^{\left(1\right)}\times_{2}\cdots \times_{M}U^{\left(M\right)}$$
where $U^{\left(m\right)}\in R^{R_{m}\times I_{m}}$ is the *m*-th factor matrix associated with the *m*-mode fiber space of X, G captures the level of interaction on each factor matrix, and $R_{m}=\mathrm{rank}\left(X_{\left(m\right)}\right)$.

Similarly, canonical polyadic decomposition [30] aims to approximate a given *M*th-order tensor $X\in R^{I_{1}\times \cdots \times I_{M}}$ into as a sum of *R* rank-one tensors, as Equation (Equation 8) shows:(8)$$X≊\sum_{r=1}^{R}\lambda_{r}X^{\left(r\right)}=\sum_{r=1}^{R}\lambda_{r}\cdot \left(u^{\left(r,1\right)}⊗\cdots ⊗u^{\left(r,M\right)}\right)$$
where $\lambda_{r}\in R$ is the *r*-th weighting term, and $u^{\left(r,m\right)}\in R^{I_{m}}$ is the *m*-mode factor vector for the *r*-th rank-one tensor $X^{\left(r\right)}$, while Equation (Equation 8) is exact iff *R* is the decomposition rank.

While MSL effectively mitigates several drawbacks related to LSL methods, it has some disadvantages. First, the intricate mathematical operations required for MSL methods very often involve high computational complexity, impacting both time and storage requirements. Moreover, MSL requires a substantial amount of data to effectively capture the intricate relationships of multilinear subspaces. Therefore, addressing these challenges is crucial to ensuring proper learning.

## 3. Problem Statement and Mathematical Definition

In this section, the problem to be addressed is formulated in natural language, outlining specific tasks related to VTS systems. Subsequently, the inherent challenges are mathematically formulated.

### 3.1. Problem Statement

Given a traffic surveillance video of τ seconds, recorded with a static camera, where multiple moving vehicles are observed, we aim to comprehensively model vehicle traffic using a multitask, multi-view learning approach. This model simultaneously addresses various tasks, such as vehicle detection, classification, and occlusion detection, each of them represented by specific views that partially describe the underlying problem. By projecting multi-view data into a unified, low-dimensional latent tensor space, which builds a new input space for the tasks, our approach should improve the model performance and provide a more comprehensive representation of different study cases, e.g., the traffic scene, compared to single-task, single-view learning models.

### 3.2. Mathematical Definition

#### 3.2.1. Multitask, Multi-View Dataset: The Input and Output Spaces

Consider a collection of *T* supervised classification tasks related to VTS systems, such as vehicle detection, classification, and occlusion detection, where, to the *t*-th task, corresponds a dataset, $D^{\left(t\right)}$, composed of $K_{t}$ *M*-view labeled instances, e.g., moving vehicles, as Equation (Equation 9) shows:(9)$$D^{\left(t\right)}=\left\{\left(x^{\left(1,1,t\right)}\cdots ,x^{\left(1,M,t\right)},y^{\left(1,t\right)}\right),\cdots ,\left(x^{\left(K_{t},1,t\right)}\cdots ,x^{\left(K_{t},M,t\right)},y^{\left(K_{t},t\right)}\right)\right\}$$
where $x^{\left(k,m,t\right)}$ is the feature vector of the *k*-th instance over the *m*-th view and *t*-th task, belonging to the feature space $X_{m}\subset R^{I_{m}}$, i.e., $x^{\left(k,m,t\right)}\in X_{m}$, the *M*-tuple $\left(x^{\left(k,1,t\right)},\cdots ,x^{\left(k,M,t\right)}\right)$ is an element of the input data space $X_{1}\times \cdots \times X_{M}$, and $y^{\left(k,t\right)}$ its corresponding true label in an output space, $Y_{t}\subset R^{O_{t}}$.

#### 3.2.2. Task Functions

For the *t*-th task, we aim to learn a multi-view classification function, $f_{t}:X_{1}\times \cdots \times X_{M}\to Y_{t}$, that predicts, with high probability, the true label ${\hat{y}}^{\left(k,t\right)}$ of the *k*-th instance, as Equation (Equation 10) shows, where $f_{t}$ belongs to some hypothesis space, $H_{t}$.
(10)$${\hat{y}}^{\left(k,t\right)}=f_{t}\left(x^{\left(k,1\right)},\cdots ,x^{\left(k,M\right)}\right)$$

Consequently, the dimension $O_{t}$ of the output space $Y_{t}$ represents the number of classes in the *t*-th learning task.

#### 3.2.3. The Parametric Model

Considering the high-dimension of the input data space, it seems reasonable to project multi-view data onto a low-dimensional latent space, S, by learning some mapping $g:X_{1}\times \cdots \times X_{M}\to S$, as Equation (Equation 11) shows:(11)$$\begin{array}{cccc} \; & \; & & Z^{\left(k\right)}=g\left(x^{\left(k,1\right)},\cdots ,x^{\left(k,M\right)}\right)\\ \; & s.t. & \; & \dim\left(Z^{\left(k\right)}\right)\leq \dim\left(X_{1}\times \cdots \times X_{M}\right) \end{array}$$
where $Z^{\left(k\right)}\in S$ is the projection of the *k*-th instance, and  $\dim\;(Z^{\left(k\right)})$ can be either unidimensional (e.g., *J*), or multidimensional (e.g., $J_{1}\times \cdots \times J_{P}$). If we need a more efficient mapping, g, a low-rank approximation function, $\hat{g}$, is required instead of g.

Let $h_{t}:S\to Y_{t}$ be the *t*-th task-specific mapping that predicts the label ${\hat{y}}^{\left(k,t\right)}$ from the *k*-th instance $Z^{\left(k\right)}$ embedded in the latent space S, as shown in Equation (Equation 12), where $h_{t}$ can be represented by, e.g., ANN, SVM, or random forest (RF) algorithms. In consequence, the function composition $h_{t}\circ g:X_{1}\times \cdots \times X_{M}\to Y_{t}$ can determine the *t*-th task function $f_{t}$.
(12)$${\hat{y}}^{\left(k,t\right)}=h_{t}\left(Z^{\left(k\right)}\right)$$

#### 3.2.4. The Optimization Problem

For a given multitask, multi-view dataset, $\left\{D^{\left(1\right)},\cdots ,D^{\left(T\right)}\right\}$, our problem can be reduced to learn simultaneously the set of functions $\left\{f_{1},\cdots ,f_{T}\right\}$ that minimizes the multi-objective empirical risk of Equation (Equation 13) [114]:(13)$$\begin{array}{cccc} \; & \min_{h_{1},\cdots ,h_{T},g} & \; & \sum_{t=1}^{T}\frac{\lambda_{t}}{K_{t}}\sum_{k=1}^{K_{t}}L_{t}\left(\left(Z^{\left(k\right)},y^{\left(k,t\right)}\right),h_{t}\right)\\ \; & s.t. & \; & Z^{\left(k\right)}=g\left(x^{\left(k,1\right)},\cdots ,x^{\left(k,M\right)}\right)\\ \; & \; & & \dim\left(Z^{\left(k\right)}\right)\leq \dim\left(X_{1}\times \cdots \times X_{M}\right) \end{array}$$
where $f_{t}=h_{t}\circ g$ belongs to some hypothesis space $H_{t}$, $L_{t}:S\times Y_{t}\times H_{t}\to R_{+}$ is the loss function related to the *t*-th task that measures the discrepancy between the true label and the predicted one, and $\lambda_{t}\in R_{+}$ is a weighting parameter, determined either statically or dynamically, which controls the relative importance of the *t*-th task.

#### 3.2.5. Objectives

The main objectives are as follows:For a multi-view input space of *M* views, to learn a mapping $g:X_{1}\times \cdots \times X_{M}\to S$, where S is a low-dimensional latent tensor space with $\dim\left(S\right)=J_{1}\times \cdots \times J_{M}$ or *J* (see Section 5.1, particularly Equation (Equation 20)).To reduce the computational complexity of g, a low-rank approximation, $\hat{g}$, needs to be learned.For a set of *T* tasks, e.g., VTS tasks, the set of task-specific functions $h_{1},\cdots ,h_{T}$ must be learned, where $h_{t}:S\to Y_{t}$, and $Y_{t}$ is the output space of the *t*-th task.To evaluate the performance of our approach, a multitask, multi-view model for the case study of VTS systems (see Section 6.2) is employed.

## 4. Vehicle Traffic Surveillance System: Multitask, Multi-View Input Space Formation

In this section, we provide a general description of several tasks associated with a typical vision-based VTS system, including background and foreground segmentation, occlusion handling, and vehicle-size classification. Together, these tasks enable the estimation of traffic parameters, such as traffic density, vehicle count, and lane occupancy, inferred from the video. Specifically, these parameters are essential for high-level ITS applications.

### 4.1. Background and Foreground Segmentation

Let $V\in Q^{H\times W\times B\times N}$ be a fourth-order tensor representing a traffic surveillance video, recorded at a FPS frame rate with a duration of τ seconds. Here, Q={0,⋯,255}, *W*, and *H* represent the image spatial dimensions, corresponding to width and height, respectively, and *B* is the dimensionality of the image spectral coordinate system, i.e., the color space in which each pixel lives, or the number of spectral bands in hyper-spectral imaging (HSI). For example, B=1 corresponds to grayscale, while B=3 corresponds to RGB color space. Finally, N=τ·FPS denotes the number of frames in the video.

From the aforementioned tensor V, it is important to note the following:The *n*th frontal slice $V_{:::n}\in Q^{H\times W\times B}$ represents the *n*th frame of the video at time $t_{n}\forall n\in \left[N\right]$.The third-mode fiber $V_{ji:n}\in Q^{B}$ denotes the (i,j)th pixel value at frame *n*, where (i,j)∈I is the pixel location belonging to the image spatial domain I=[W]×[H].Each pixel value is quantized using *D* bits per spectral band. For simplicity, here, we assume the 8-bit grayscale color space Q={0,⋯,255}, i.e., B=1, but it can be extended to other color spaces. Consequently, dim(V) reduces to H×W×N.Every (i,j)th pixel value can be modeled as a discrete random variable, $X_{ij}$, with a probability mass function (pmf), denoted as $P(X_{ij}=x)$, where x∈Q.For any observation time, $\tau_{o}<\tau$, the pmf of any pixel can be estimated, denoted as $\hat{P}\left(X_{ij}=x\right)$.

Then, tensor V can be decomposed as Equation (Equation 14) shows and Figure 1 illustrates:(14)$$V=B⊙\bar{M}+F$$
where $B\in Q^{H\times W\times N}$ is called the background tensor, $F\in Q^{H\times W\times N}$ is the foreground tensor, and $M\in B^{H\times W\times N}$ is the binary mask of the foreground tensor, whose (j,i,n)-th entry $m_{\mathrm{jin}}=1$ if the (i,j)th pixel value $v_{\subset }\mathrm{jin}$ of V at frame *n* is part of the foreground tensor F; otherwise $m_{\mathrm{jin}}=0$, $\bar{M}\in B^{H\times W\times N}$ the complement of M, and F can be obtained from the Hadamard product V⊙M.

### 4.2. Blob Formation

After decomposing V into the background and foreground tensors, various moving objects, including vehicles, pedestrians, and cyclists, can be extracted by analyzing F or its mask, M. One such technique is called connected components analysis (CCA) [115,116]. CCA recursively searches at every *n*th frontal slice $M_{::n}$ for connected pixel regions (see Definition 12), referred to in the literature as binary large objects (blobs), which can contain pixels associated with moving objects.

**Definition** **12**(Blob)**.** *A blob, denoted as S, is a set of pixel locations connected by a specified connectivity criterion (e.g., four-connectivity or eight-connectivity* [117]*). Specifically, a pixel located at (i,j)∈I belongs to blob S if there exists another pixel location, $(i^{\prime },j^{\prime })\in S$, such that the connectivity criterion is met, as Equation (Equation 15) shows:*
(15)$$S=\{\left(i,j\right)\in I|\exists \left(i^{\prime },j^{\prime }\right)\in S:\left(i^{\prime },j^{\prime }\right)\neq \left(i,j\right)\wedge d_{IP}\left(\left(i,j\right),\left(i^{\prime },j^{\prime }\right)\right)=\delta \}$$
*where $d_{IP}:I\times I\to R$ is an inter-pixel distance that establishes the connectivity criterion given some threshold value, δ∈R, and S⊆I.*

For every blob $S^{\left(n\right)}$ detected at frame *n*, a blob mask, $S^{\left(n\right)}\in B^{H\times W}$, can be formed whose entries are given by Equation (Equation 16). Note that the pixel values of blob $S^{\left(n\right)}$ can be obtained from the product $\left(F_{::n}⊙S^{\left(n\right)}\right)\in Q^{H\times W}$.
(16)$$s_{ji}^{\left(n\right)}=\left\{\begin{array}{cc} 1, & \left(i,j\right)\in S^{\left(n\right)}\\ 0, & \mathrm{otherwise} \end{array}\right.$$

### 4.3. Vehicle Feature Extraction and Selection

Feature extraction can be considered a mapping, $\zeta :Q^{H\times W}\to X$, that transforms a given blob, *S*, into a low-dimensional point, x∈X, called the feature vector, as shown in Equation (Equation 17):(17)$$x=\zeta \left(F_{::n}⊙S\right)$$
where X, called the feature space, captures specific aspects of blobs $S^{\left(n\right)}$, e.g., color, shape, or texture.

The image moments (IMs) are a classical hand-crafted feature extractor that provides information about the spatial distribution, shape, and intensity of a blob image. Typical features extracted via the IM include centroid, area, orientation, and eccentricity. Formally, the (p,q)-th raw IM for blob *S* is given by the bilinear map of Equation (Equation 18):(18)$$x_{pq}=\sum_{j=1}^{W}\sum_{i=1}^{H}j^{p}i^{q}b_{ij}=B\times_{1}\zeta \times_{2}\eta$$
where $\zeta \in R^{H}$, $\eta \in R^{W}$ are vectors whose *i*-th and *j*-th entries are $\zeta_{i}=i^{p}$ and $\eta_{j}=j^{q}$, respectively.

### 4.4. Vehicle Occlusion Task

Assuming there are $V_{n}$ vehicles on the road at the *n*-th frame, each associated with a specific blob $S^{\left(n,v\right)}$, let $B^{\left(n\right)}$ denote the set of these blobs, and let ${\tilde{B}}^{\left(n\right)}={\left\{{\tilde{S}}^{\left(n,u\right)}\right\}}_{u=1}^{U_{n}}$ be the set of blobs detected via CCA in the *n*-th frame, where $U_{n}\leq V_{n}$. The *v*-th vehicle, with blob $S^{\left(n,v\right)}\in B^{\left(n\right)}$, is occluded by the *u*-th detected blob ${\tilde{S}}^{\left(n,v\right)}\in {\tilde{B}}^{\left(n\right)}$ if any of the conditions in Equation (Equation 19) are met.
(19)$$\left\{\begin{array}{cc} S^{\left(n,v_{1}\right)}\cap {\tilde{S}}^{\left(n,v_{2}\right)}\neq \emptyset & S^{\left(n,v\right)}\;\mathrm{and}\;{\tilde{S}}^{\left(n,u\right)}\;\mathrm{are}\;\mathrm{occluded}\\ S^{\left(n,v\right)}\cap {\tilde{S}}^{\left(n,u\right)}={\tilde{S}}^{\left(n,u\right)} & S^{\left(n,u\right)}\;\mathrm{is}\;\mathrm{totally}\;\mathrm{occluded}\;\mathrm{by}\;{\tilde{S}}^{\left(n,v\right)}\\ \left(S^{\left(n,v\right)}\cap {\tilde{S}}^{\left(n,u\right)}\neq \emptyset \right)\wedge \left(S^{\left(n,v\right)}\cap {\tilde{S}}^{\left(n,u\right)}\neq S^{\left(n,v\right)}\right) & S^{\left(n,v\right)}\;\mathrm{is}\;\mathrm{partially}\;\mathrm{occluded} \end{array}\right.$$

Given the set of detected blobs ${\tilde{B}}^{\left(n\right)}$ in the *n*-th frame, the vehicle occlusion detection aims to predict, with high probability, a set of $W_{n}\leq U_{n}$ blobs ${\hat{B}}^{\left(n\right)}=\left\{{\hat{S}}^{\left(n,1\right)},\cdots ,{\hat{S}}^{\left(n,W_{n}\right)}\right\}$, each containing more than one vehicle. To achieve this, an occlusion feature space, $X_{1}\subset R^{I_{1}}$, is constructed using a feature extraction mapping $\zeta_{1}:Q^{W\times H}\to X_{1}$ to capture the vehicle occlusion patterns. In this space, every detected blob, ${\tilde{S}}^{\left(n,u\right)}$, is represented by an $I_{1}$-dimensional feature vector $x^{\left(n,u\right)}\in X_{1}$. Assuming occlusions are only composed of partially observed vehicles, a classification function, $f_{1}:X_{1}\to \left\{0,1\right\}$, can be built to predict whether a detected blob, ${\tilde{S}}^{\left(n,u\right)}$, has more than one vehicle.

### 4.5. Vehicle Classification Task

Given a set of vehicle-size labels (e.g., small,midsize,large) represented in a vector space, $Y_{2}\subset R^{O_{2}}$, called the output space, the vehicle classification task aims to predict, with high probability, the true label $y^{\left(n,u\right)}\in Y_{2}$ for an unseen vehicle blob instance, ${\tilde{S}}^{\left(n,u\right)}\in {\tilde{B}}^{\left(n\right)}$, at frame *n*. First, each blob, ${\tilde{S}}^{\left(n,u\right)}$, is mapped into some feature space, $X_{2}\subset R^{I_{2}}$, using a feature extraction mapping, $\zeta_{2}:Q^{W\times H}\to X_{2}$, constructed to explain the vehicle-size patterns. From this space, a feature vector, $x^{\left(n,u\right)}$, associated with ${\tilde{S}}^{\left(n,u\right)}$, is derived. Then, a classification function, $f_{2}:X_{2}\to Y_{2}$, can be built to predict the label of a vehicle blob instance, ${\tilde{S}}^{\left(n,w\right)}$.

## 5. A Multi-View Data Tensor Fusion Layer and the Connection Between the Einstein and Hadamard Products

In this section, the concept of a multi-view data tensor fusion (MV-DTF) layer and its connection with Einstein and Hadamard products are introduced. Basically, MV-DTF is a form of an FC layer for multi-view data; i.e., it is an affine function, but instead of using a linear map, our layer employs a multilinear map to encode the interactions across views. Additionally, a low-rank approximation for the MV-DTF layer is also proposed to reduce its computational complexity.

### 5.1. Multi-View Tensor Data Fusion Layer: The Mapping *g* as an Einstein Product

Inspired by previous works [36,75,76], we restrict the function space of the MV-DTF layer to the affine functions characterized by a multilinear map, $T:X_{1}\times \cdots \times X_{M}\to S$, followed by a translation and, possibly, a non-linear map, σ, as Equation (Equation 20) shows:(20)$$Z^{\left(k\right)}=g\left(x^{\left(k,1\right)},\cdots ,x^{\left(k,M\right)}\right)=\sigma \left(T\left(x^{\left(k,1\right)},\cdots ,x^{\left(k,M\right)}\right)+B\right)$$
where g is the MV-DTF layer, the *P*-order tensor $Z^{\left(k\right)}\in S$ is the projection of the *k*-th instance $\left(x^{\left(k,1\right)},\cdots ,x^{\left(k,M\right)}\right)$ onto the latent tensor space S, called the fused tensor, with dimension $J_{1}\times \cdots \times J_{P}$, $B\in R^{J_{1}\times \cdots \times J_{P}}$ is the translational term, formally called bias, and the mapping $\sigma :S\to R^{J_{1}\times \cdots \times J_{P}}$.

Definition 13 specifies how a multilinear map can be represented using coordinate systems, and from this representation, a tensor can be induced for every multilinear map.

**Definition** **13**(Coordinate representation of a multilinear map [101])**.** *Let $V_{1},\ldots ,V_{M},$ and W be real vector spaces, where $\dim\left(V_{m}\right)=I_{m}$ for all m∈[M], and dim(W)=J. Let $\left\{e^{\left(1\right)},\cdots ,e^{\left(J\right)}\right\}$ be the standard basis for W. And let $T:V_{1}\times \cdots \times V_{M}\to W$ be a multilinear map. Given an ordered M-tuple $\left(x^{\left(1\right)},\cdots ,x^{\left(M\right)}\right)\in V_{1}\times \cdots \times V_{M}$, where $x^{\left(m\right)}\in V_{m}$, the map $T(x^{\left(1\right)},\cdots ,x^{\left(M\right)})$ is completely determined by a linear combination of basis vectors $e^{\left(1\right)},\cdots ,e^{\left(J\right)}$ and scalars $\left\{\alpha_{ji_{1}\cdots i_{M}}\in R|i_{1}\in \left[I_{1}\right],\cdots ,i_{M}\in \left[I_{M}\right],j\in \left[J\right]\right\}$, as Equation (Equation 21) shows.*
(21)$$T\left(x^{\left(1\right)},\cdots ,x^{\left(M\right)}\right)=\sum_{i_{1}=1}^{I_{1}}\cdots \sum_{i_{M}=1}^{I_{M}}\sum_{j=1}^{J}x_{i_{1}}^{\left(1\right)}\cdots x_{i_{M}}^{\left(M\right)}\alpha_{ji_{1}\cdots i_{M}}e^{\left(j\right)}$$
*The collection of scalars can then be arranged into an (M+1)th-order tensor, denoted as $A\in R^{J\times I_{1}\times \cdots \times I_{M}}$, which determines T, and whose $\left(j,i_{1},\cdots ,i_{M}\right)$-th entry $a_{ji_{1}\cdots i_{M}}$ corresponds with $\alpha_{ji_{1}\cdots i_{M}}$.*


Next, Definition 14 establishes a connection between the Einstein product and multilinear maps via the universal property of multilinear maps (see Definition 2).

**Definition** **14**.*Let $x^{\left(1\right)},\cdots ,x^{\left(M\right)}$ be a set of vectors, where $x^{\left(m\right)}\in R^{I_{m}}$ for all m∈[M]. And let $T:R^{I_{1}}\times \cdots \times R^{I_{M}}\to R^{J}$ be the multilinear map induced via the tensor $A\in R^{J\times I_{1}\times \cdots \times I_{M}}$, and $\pi :R^{I_{1}}\times \cdots \times R^{I_{M}}\to R^{I_{1}}⊗\cdots ⊗R^{I_{M}}$ is the multilinear map associated with the tensor product $R^{I_{1}}⊗\cdots ⊗R^{I_{M}}$. For $X=\pi \left(x^{\left(1\right)},\cdots ,x^{\left(M\right)}\right)\in R^{I_{1}}⊗\cdots ⊗R^{I_{M}}$, the Einstein product $A{⊛}_{M}X$ can be understood as a linear map, $\Phi :R^{I_{1}}⊗\cdots ⊗R^{I_{M}}\to R^{J}$. Then, T and* Φ *are related by the universal property of multilinear maps, as Equation (Equation 22) shows.*
(22)$$T\left(x^{\left(1\right)},\cdots ,x^{\left(M\right)}\right)=\left(\Phi \circ \pi \right)\left(x^{\left(1\right)},\cdots ,x^{\left(M\right)}\right)=\Phi \left(X\right)=A{⊛}_{M}X$$

For the multilinear map $T:X_{1}\times \cdots \times X_{M}\to S$ in Equation (Equation 20), Definition 13 ensures the existence of a tensor, $A\in R^{J_{1}\times \cdots \times J_{P}\times I_{1}\times \cdots \times I_{M}}$, that determines T, and Definition 14 provides the associated linear map $\Phi :R^{I_{1}}⊗\cdots ⊗R^{I_{M}}\to S$ of T. From the above definitions, Equation (Equation 20) can be rewritten in tensor form as Equation (Equation 23) shows, where $X^{\left(k\right)}=x^{\left(k,1\right)}⊗\cdots ⊗x^{\left(k,M\right)}\in R^{I_{1}\times \cdots \times I_{M}}$ is a rank-one tensor resulted from the tensor product of the *M* view vectors associated with the *k*-th instance.
(23)$$Z^{\left(k\right)}=g\left(x^{\left(k,1\right)},\cdots ,x^{\left(k,M\right)}\right)=\sigma \left(A{⊛}_{M}X^{\left(k\right)}+B\right)$$

Note that Equation (Equation 23) represents a differentiable expression with respect to tensors A and B. Consequently, their values can be learned using optimization algorithms such as stochastic gradient descent (SGD), where the number of parameters to learn, denoted as *L*, corresponds with the number of entries of tensors A and B, as Equation (Equation 24) shows. Note that *L* scales exponentially with the number of views, *M*, and the order *P* of S. Specifically, for $I_{m}=J_{p}=I\;\forall m\in \left[M\right],p\in \left[P\right]$, *L* is reduced to $L=I^{P}\left(1+I^{M}\right)≃I^{M+P}$. This exponential growth can lead to computational challenges while increasing the risk of overfitting due to the induced curse of dimensionality [118,119,120]; i.e., the number of samples needed to train a model grows exponentially with its dimension.
(24)$$L=\left(\prod_{m=1}^{M}I_{m}\right)\cdot \left(\prod_{p=1}^{P}J_{p}\right)+\prod_{p=1}^{P}J_{p}=\prod_{p=1}^{P}J_{p}\left(\prod_{m=1}^{M}I_{m}+1\right)$$

### 5.2. Hadamard Products of Einstein Products and Low-Rank Approximation Mapping $\hat{g}$

Low-rank approximation is a well-known technique that not only allows for reducing model parameter storage requirements but also helps in alleviating the computational burden of neural network models [81,82,85,86,87,88,89,121]. Based on these facts, in this work, we explore a CPD-based low-rank structure, illustrated in Figure 2, to overcome the curse of dimensionality induced via the MV-DTF layer. This structure helps reduce the number of parameters required for the MV-DTF layer, and it is computationally more efficient (see Proposition 1). But before presenting this structure, the concept of Hadamard factor tensors is first introduced in Definition 15.

**Definition** **15**(Hadamard factor tensors)**.** *Let $A\in R^{J_{1}\times \cdots \times J_{P}\times I_{1}\times \cdots \times I_{M}}$ be a (P+M)-order tensor, whose $\left(j_{1},\cdots ,j_{P}\right)$-th subtensor results from fixing every index but the last M modes; i.e., $A_{j_{1}\cdots j_{P}:\cdots :}\in R^{I_{1}\times \cdots \times I_{M}}$ for all $j_{p}\in \left[J_{p}\right]$ and p∈[P] can be approximated as a rank-$R^{\left(j_{1},\cdots ,j_{P}\right)}$ tensor using the CPD, as Equation (Equation 25) shows:*
(25)$$A_{j_{1}\cdots j_{P}:\cdots :}≊\sum_{r=1}^{R^{\left(j_{1},\cdots ,j_{P}\right)}}v^{\left(j_{1},\cdots ,j_{P},r,1\right)}⊗\cdots ⊗v^{\left(j_{1},\cdots ,j_{P},r,M\right)}$$
*where the number of subtensors in A corresponds to the dimension of the latent space S; i.e., $J_{1}\times \cdots \times J_{P}$, each $\left(j_{1},\cdots ,j_{P}\right)$-th subtensor $Aj_{1}\cdots j_{P}:\cdots :$ has a specific rank, $R^{\left(j_{1},\cdots ,j_{P}\right)}\in N$, which can be different across subtensors, and for $v^{\left(j_{1},\cdots ,j_{P},r,m\right)}\in R^{I_{m}}$, known as the m-mode factor vector, the superscripts $j_{1},\cdots ,j_{P}$ identify the $\left(j_{1},\cdots ,j_{P}\right)$-th subtensor to which it corresponds, $r\in [R^{\left(j_{1},\cdots ,j_{P}\right)}]$ identifies its associated r-th rank-one tensor in the CPD, and m∈[M] its mode. Then, the set of factor vectors along the m-mode can be rearranged into a (P+2)-order tensor, $U^{\left(m\right)}\in R^{J_{1}\times \cdots \times J_{P}\times R\times I_{m}}$, here called the m-mode Hadamard factor tensor, whose (P+2)-mode fibers $U_{j_{1}\cdots j_{P}r:}^{\left(m\right)}\in R^{I_{m}}$ are given by Equation (Equation 26):*
(26)$$U_{j_{1}\cdots j_{P}r:}^{\left(m\right)}=\left\{\begin{array}{cc} v^{\left(j_{1},\cdots ,j_{P},r,m\right)}, & r\leq R^{\left(j_{1},\cdots ,j_{P}\right)}\\ 0, & r>R^{\left(j_{1},\cdots ,j_{P}\right)} \end{array}\right.$$
*where $0\in R^{I_{m}}$ is the zero vector, and $R=\max R^{\left(j_{1},\cdots ,j_{P}\right)}$ is the maximum rank across subtensors, employed to avoid inconsistencies due to different rank values between subtensors.*

Figure 2 illustrates the concept of Hadamard-factor tensors for the multilinear map $T:R^{5}\times R^{3}\to R^{3}$ with associated tensor $A\in R^{3\times 5\times 3}$. Here, there is a two-view data (M=2) with dimensions $I_{1}=5$, and $I_{2}=3$, respectively; the order and dimension of the latent tensor space are P=1 and $J_{1}=J=3$, and hence, there are three subtensors, $A_{1::}$, $A_{2::},A_{3::}\in R^{5\times 3}$, associated with the tensor A. For subtensor $A_{1::}$, its rank is $R^{\left(1\right)}=3$; hence, $A_{1::}=v^{\left(1,1,1\right)}⊗v^{\left(1,1,2\right)}+v^{\left(1,2,1\right)}⊗v^{\left(1,2,2\right)}+v^{\left(1,3,1\right)}⊗v^{\left(1,3,2\right)}$ for subtensor $A_{2::}$, $R^{\left(2\right)}=1$, i.e., $A_{2::}=v^{\left(2,1,1\right)}⊗v^{\left(2,1,2\right)}$, while for subtensor $A_{3::}$, $R^{\left(3\right)}=2$, and $A_{3::}=v^{\left(3,1,1\right)}⊗v^{\left(3,1,2\right)}+v^{\left(3,2,1\right)}⊗v^{\left(3,2,2\right)}$. From these vectors, two Hadamard factor tensors, $U^{\left(1\right)}\in R^{3\times 3\times 5}$ and $U^{\left(2\right)}\in R^{3\times 3\times 3}$, can be constructed, corresponding to the first and second views, respectively. The second-mode dimension of these tensors corresponds to the greatest subtensor rank, i.e., $R=\max R^{\left(j\right)}=R^{\left(1\right)}=3$, to avoid heterogeneous rank values across subtensors. Hence, the second and third subtensors incorporate two and one additional zero vectors, respectively, as Figure 2 shows.

Proposition 1 presents the primary result of this work, i.e., the mathematical relationship between Einstein and Hadamard products. To the best of our knowledge, this relationship is not known.

**Proposition** **1.**
*Let $X=x^{\left(1\right)}⊗\cdots ⊗x^{\left(M\right)}\in R^{I_{1}\times \cdots \times I_{M}}$ be a rank-one tensor, where $x^{\left(m\right)}\in R^{I_{m}}$ for all m∈[M]. And let $A\in R^{J_{1}\times \cdots \times J_{P}\times I_{1}\times \cdots \times I_{M}}$ be a (P+M)-order tensor induced via the multilinear map $T:R^{I_{1}}\times \cdots \times R^{I_{M}}\to R^{J_{1}\times \cdots \times J_{P}}$, which can be decomposed into a set of M factor tensors $U^{\left(1\right)},\ldots ,U^{\left(M\right)}$ for a given rank, $R\leq \max\mathrm{rank}(A_{j_{1}\cdots j_{P}:\cdots :})$, where $U^{\left(m\right)}\in R^{J_{1}\times \cdots \times J_{P}\times R\times I_{m}}$ for all m∈[M]. Then, $T(x^{\left(1\right)},\cdots ,x^{\left(M\right)})$ can be approximated by a sum of R Hadamard products of Einstein products, as Equation (Equation 27) shows:*

(27)
$$T\left(x^{\left(1\right)},\cdots ,x^{\left(M\right)}\right)=A{⊛}_{M}X≊\sum_{r=1}^{R}\overset{M}{\underset{m=1}{⨀}}\left(U_{:\cdots :r:}^{\left(m\right)}{⊛}_{1}x^{\left(m\right)}\right)=\left[\overset{M}{\underset{m=1}{⨀}}\left(U^{\left(m\right)}{⊛}_{1}x^{\left(m\right)}\right)\right]{⊛}_{1}1^{R}$$

*where $U^{\left(m\right)}\in R^{J_{1}\times \cdots \times J_{P}\times R\times I_{m}}$ is the m-mode Hadamard factor tensor. A vector of all ones is denoted as $1^{R}\in R^{R}$. And $\overset{M}{\underset{m=1}{⨀}}\left(U^{\left(m\right)}{⊛}_{1}x^{\left(m\right)}\right)=\left(U^{\left(1\right)}{⊛}_{1}x^{\left(1\right)}\right)⊙\cdots ⊙\left(U^{\left(M\right)}{⊛}_{1}x^{\left(M\right)}\right)$.*


**Proof.** In Appendix A.    □

By leveraging Proposition 1 for tensor A in Equation (Equation 23), the MV-DTF layer g can be approximated through a more efficient low-rank mapping $\hat{g}:R^{I_{1}}\times \cdots \times R^{I_{M}}\to R^{J_{1}\times \cdots \times J_{P}}$, called the low-rank multi-view data tensor fusion (LRMV-DTF) layer, defined in Equation (Equation 28), where the *m*-mode factor tensor $U^{\left(m\right)}\in R^{J_{1}\times \cdots \times J_{M}\times I_{m}\times R}$, associated with the *m*-th view, contributes to building every *k*-th fused tensor $Z^{\left(k\right)}$.
(28)$$Z^{\left(k\right)}≊\hat{g}\left(x^{\left(k,1\right)},\cdots ,x^{\left(k,M\right)}\right)=\sigma \left(\left[\overset{M}{\underset{m=1}{⨀}}\left(U^{\left(m\right)}{⊛}_{1}x^{\left(k,m\right)}\right)\right]{⊛}_{1}1+B\right)$$

From this approximation, the number of parameters required for the LRMV-DTF layer, denoted as $\hat{L}$, is provided in Equation (Equation 29). Note that the product of the $I_{m}$-dimensions related to the views in *L* (Equation (Equation 24)) has been replaced with a summation, which yields fewer parameters to learn compared to those in the MV-DTF layer, reducing the risk of overfitting.
(29)$$\hat{L}=\prod_{p=1}^{P}J_{p}\cdot \left(R\sum_{m=1}^{M}I_{m}+1\right)$$

An illustration of our layers is shown in Figure 3a (MV-DTF), and Figure 3b (LRMV-DTF). Here, the number of views M=2, and their dimensions $I_{1}=3$, and $I_{2}=5$ respectively. The order of the latent space is P=1, and its dimension dim(S)=J=4. Consequently, the multilinear map is $T:R^{3}\times R^{5}\to R^{4}$, with associated tensor $A\in R^{4\times 3\times 5}$, and bias $b\in R^{4}$. However, vector b is fixed to zero $0\in R^{4}$ for simplicity. For low-rank approximation, the rank of the (j)-th subtensor is $R^{\left(j\right)}=R=2\;\forall j\in \left[4\right]$. Hence, according to Definition 15, tensor A can be decomposed into two factor tensors: $U^{\left(1\right)}\in R^{4\times 2\times 3}$, and $U^{\left(2\right)}\in R^{4\times 2\times 5}$, associated with the first and second views, respectively.

This relationship between the Einstein and Hadamard product enables a rank-*R* CPD for every subtensor $\left(j_{1},\cdots ,j_{P}\right)$ of tensor A and, consequently, a low-rank approximation.

### 5.3. Dimension J or $J_{1}\times \cdots \times J_{P}$, Order P of the Latent Space S, and the Rank R: The Hyperparameters of the MV-DTF and LRMV-DTF Layers

The proposed layers introduce three hyperparameters to tune: the order *P* and the dimension $J_{1}\times \cdots \times J_{P}$ of S, and the rank value *R*:**Latent space dimension**: It determines the expressiveness of the latent space to capture complex patterns across views. High-dimensional spaces enhance expressiveness but also increase the risk of overfitting, while low-dimensional spaces reduce expressiveness but mitigate the risk of overfitting.**Latent space order**: It is determined by the architecture of the ANN. For instance, in multi-layer perceptron (MLP) architectures, the input space dimension is unidimensional, e.g., dim(S)=J; hence, P=1. In contrast, the input space of CNNs is multidimensional; thereby, P≥2.**Rank**: It determines the computational complexity of the MV-DTF layer. For low rank values on subtensors $A_{j_{1}\cdots j_{P}:\cdots :}$ of A, the number of parameters to learn can be reduced, but it may not capture complex interactions across views effectively, limiting the model performance. Conversely, high rank values increase the capacity to learn complex patterns in data, but they may lead to overfitting.

### 5.4. MV-DTF and LRMV-DTF on Neural Network Architectures: The Mapping Set $\left\{h_{1},\cdots ,h_{T}\right\}$

According to the desired level of fusion [16], two primary configurations can be employed where our data fusion layer can be incorporated in an ANN architecture:**Feature extraction**: The MV-DTF layer *g* can be integrated into an ANN to map the multi-view input space $X_{1}\times \cdots \times X_{M}$ into some latent space, S, for multi-view feature extraction; see Figure 4a,c. Here, both the order *P* and dimension $J_{1}\times \cdots \times J_{P}$ of S must correspond with the order and dimension of the input layer in the architecture of the ANN.**Multilinear regression**: The MV-DTF layer performs multilinear regression to capture the multilinear relationships between the multi-view latent space $U_{1}\times \cdots \times U_{M}$ and the output space Y for single-task learning (see Figure 4b). Here, $U_{m}$ is the *m*-th single-view latent space obtained from the mapping $\zeta_{m}:X_{m}\to U_{m}$, where $X_{m}$ is the *m*-th single-view input space. Consequently, the dimension and order of the latent space must correspond with those of the output space.

## 6. Results and Discussion

### 6.1. Dataset Description

To test the effectiveness of the proposed MV-DTF layer, we conduct experiments on four real-world traffic surveillance videos, encompassing more than 50,000 frames of footage with a resolution of 420 × 240 pixels and recorded at a frame rate of 25 FPS (accessible via [122]). Sample images from each test video can be observed in Figure 5, while technical details are provided in Table 3.

A collection of over 92,000 images of vehicles was then extracted from the test videos using the background and foreground method. Each image has been manually labeled for two tasks (T=2): (1) occlusion detection, where vehicles are categorized as occluded or non-occluded (labeled to as 1 and 0, respectively), and (2) vehicle-size classification, where non-occluded vehicles are categorized as small (S), midsize (M), large (L), or very large (XL), with labels 1 to 4. Next, one-hot encoding was used to represent the class labels of each task. Consequently, the output spaces for the classification and occlusion detection tasks become $Y_{1}\subset B^{4}$, and $Y_{2}\subset B^{2}$, respectively, i.e., $O_{1}=4$ and $O_{2}=2$.

In addition, three subsets of image moment-based features were extracted and normalized for each vehicle image: (1) a 4D feature space, $X_{1}\subset R^{4}$ (i.e., $I_{1}=4$), consisting of the vehicle blob solidity, orientation, eccentricity, and compactness features; (2) a 3D feature space, $X_{2}\subset R^{3}$ (i.e., $I_{2}=3$), encompassing the vehicle’s width, area, and aspect ratio; and (3) a 2D feature space, $X_{3}\subset R^{2}$, representing the vehicle centroid coordinates. Together, the three feature spaces form a three-view input space $X_{1}\times X_{2}\times X_{3}$ of dimension 4×3×2; i.e., the number of views is M=3, and $I_{1}=4$, $I_{2}=3$, and $I_{3}=2$ are the dimensions of each feature space.

As a result, two datasets, denoted as $D^{\left(1\right)}$ and $D^{\left(2\right)}$, were created from the test videos, where $D^{\left(1\right)}$ corresponds to the occlusion detection task and $D^{\left(2\right)}$ to the vehicle-size classification task. Both datasets encompass vehicle instances represented in a three-view feature space, available in [123]. Table 4 and Table 5 provide a summary of our datasets, detailing the distribution of images across the occlusion and vehicle-size classification tasks.

### 6.2. The Multitask, Multi-View Model Architecture and Training

#### 6.2.1. The Multitask, Multi-View Model Architecture

To learn the two tasks, a multitask, multi-view ANN model based on the MLP architecture was employed. The  proposed model is structured in four main stages (see Figure 6): (1) hand-crafted feature extractors (shown in red), (2) an MV-DTF/LRMV-DTF layer (in green), (3) the neck (in yellow), and (4) the task-specific heads (in blue). Stages 1 and 2 form the backbone of the model, serving as a feature extractor to capture both low-level and high-level features from the raw data. Stage 3 refines the features extracted from the backbone. Finally, stage 4 performs prediction or inference. In addition, dropout is applied at the end of each stage to reduce the risk of overfitting and enhance the model’s generalization.

The MV-DTF/LRMV-DTF layer provides the mapping $T:X_{1}\times X_{2}\times X_{3}\to S$, where $\dim(X_{1})=4$, $\dim(X_{2})=3$, and $\dim(X_{3})=2$. The order *P* of the latent space S is fixed to one, i.e., P=1 without loss of generality, which simplifies its dimension $J_{1}\times \cdots \times J_{P}$ to *J*, a hyperparameter to tune. Therefore, the parameters of the MV-DTF/LRMV-DTF layer are either the tensor $A\in R^{J\times 4\times 3\times 2}$, or the associated Hadamard factor tensors $U^{\left(1\right)}\in R^{J\times R\times 4},U^{\left(2\right)}\in R^{J\times R\times 3}$, and $U^{\left(3\right)}\in R^{J\times R\times 2}$, where the rank *R* is a hyperparameter to tune, along with the bias tensor $B\in R^{J}$. Consequently, Equation (Equation 28) is reduced to Equation (Equation 30) (Equation (Equation 30) holds when $R^{\left(j\right)}$ is the tensor decomposition rank for all j∈[J]):(30)$$z^{\left(k\right)}=\sigma \left(A{⊛}_{3}X^{\left(k\right)}+b\right)≊\sigma \left(\overset{3}{\underset{m=1}{⨀}}\left(U^{\left(m\right)}{⊛}_{1}x^{\left(k,m\right)}\right){⊛}_{1}1+b\right)$$
where the fused tensor $Z^{\left(k\right)}$ and bias B are transformed to the vectors $z^{\left(k\right)}\in R^{J}$ and $b\in R^{J}$, respectively, and $X^{\left(k\right)}=x^{\left(k,1\right)}⊗x^{\left(k,2\right)}⊗x^{\left(k,3\right)}\in R^{4\times 3\times 2}$.

To solve this problem (see Section 3), the multi-objective optimization defined in Equation (Equation 13) is employed with T=2 and M=3, where $h_{1}$ and $h_{2}$ are the task-specific occlusion detection and vehicle classification functions, $g:X_{1}\times X_{2}\times X_{3}\to S$ can be either the MV-DTF or LRMV-DTF layer, $L_{1}$ and $L_{2}$ are the binary cross-entropy and multiclass cross-entropy loss (see Definitions 16 and 17, respectively) for the above tasks, and the task importance weighting hyperparameters $\lambda_{1}=0.4$ and $\lambda_{2}=0.6$ were selected from a finite set of values through cross-validation, a technique often employed by other authors [69].

**Definition** **16**(Binary cross-entropy (BCE))**.** *Let y∈B be the true label of an instance, and let $\hat{y}\in \left[0,1\right]$ be the predicted probability for the positive class. The BCE between y and $\hat{y}$ is given by the following:*
(31)$$L_{1}\left(\hat{y},y\right)=-\left[y\log\left(\hat{y}\right)+\left(1-y\right)\log\left(1-\hat{y}\right)\right]$$

**Definition** **17**(Multiclass cross entropy (MCE))**.** *Let $y\in B^{C}$ be the true label of an instance, related to some multi-classification problem with C classes, encoded in one-hot format. And let $\hat{y}\in R^{C}$ be the predicted probability vector, where ${\hat{y}}_{c}$ is the probability that the instance belongs to the c-th class. The MCE between y and $\hat{y}$ is given by the following:*
(32)$$L_{2}\left(\hat{y},y\right)=-\sum_{c=1}^{C}y_{c}\log\left({\hat{y}}_{c}\right)$$

#### 6.2.2. Training and Validation

From the total number of tracked vehicles in Table 3, 45% of them were selected from the four test videos via stratified random sampling for training and validation purposes. Including all temporal instances of a particular vehicle can cause data leakage; i.e., the model may learn specific patterns from highly correlated temporal samples, resulting in reduced generalization to unseen vehicles. To prevent this, uncorrelated temporal instances were only considered for each selected vehicle.

Vehicles from the 45% subset, along with their uncorrelated temporal instances, were partitioned into two sets: (1) the training set $D{tr}^{\left(t\right)}$, containing the 30% of vehicles and their temporal instances; and (2) the validation set $D{va}^{\left(t\right)}$, with 15% of the vehicles and their instances, where superscript *t* indexes the task-specific dataset (i.e., t=1 for vehicle-size classification and t=2 for occlusion detection). The remaining 55% of vehicles and their instances comprise the testing set, denoted as $D_{te}^{\left(t\right)}$.

Adaptive moment estimation [124] was employed to optimize the parameters of our model. Training was performed for a maximum of 200 epochs, with an early stopping scheme to avoid overfitting by halting training when performance on $D_{va}^{\left(t\right)}$ no longer improved. The training strategy for our multitask, multi-view model is shown in Algorithm 1, where $F_{tr}^{\left(t,b\right)}\subset D_{tr}^{\left(t\right)}$ is a mutually exclusive batch of the *t*-th task, i.e., $F_{tr}^{\left(t,b\right)}\cap F_{tr}^{\left(t,q\right)}=\emptyset$ for b≠q, with b,q∈[K], and *K* is the number of batches.
**Algorithm 1** Training scheme.1:Initialize all weights and biases of the network randomly.2:**for** i=1 to *K* **do**3:   Optimize the model over batch $F_{tr}^{\left(1,i\right)}$ to minimize the loss for task 1.4:   Optimize the model over batch $F_{tr}^{\left(2,i\right)}$ to minimize the loss for task 2.5:**end for**

All experiments were conducted and implemented in Python 3.10 and the PyTorch framework on a computer equipped with an Intel Core i7 processor running at 2.2 GHz. To accelerate the processing time, an NVIDIA GTX 1050 TI GPU was employed.

### 6.3. Performance Evaluation Metrics

In this work, we evaluate the performance of the proposed multitask, multi-view model using six main metrics: accuracy (ACC), F1-measure (F1), geometric mean (GM), normalized Matthews correlation coefficient (MCCn), and normalized Bookmaker informedness (BMn), as detailed in Table 6 (see details of these metrics in [125]). For binary classification, where vehicle instances are categorized into two classes—positive and negative—the performance metrics were directly derived from the entries of a 2×2 confusion matrix (CM), characterized by true positives (TPs), false negatives (FNs), false positives (FPs), and true negatives (TNs). In multiclass classification with C>2 classes, the notions of TP, FN, FP, and TN are less straightforward than in binary classification, as the confusion matrix becomes a C×C matrix whose (i,j)-th entry represents the number of samples that truly belong to the i-th class but were classified as the j-th class. In order to derive the performance metrics, a one-vs.-rest approach is typically employed to reduce the multiclass CM into *C* binary CMs, where the *c*-th matrix is formed by treating the *c*-th class as positive and the rest as the negative class [125,126]. Figure 7 illustrates the CM notion for binary classification (Figure 7a) and multiclass classification with C=4 classes (Figure 7b), which were obtained from the mean values of runs.

In Table 6, *C* denotes the number of classes of interest, *M* is the number of classified instances, $\mathrm{CM}\in R^{C\times C}$ is a multiclass CM, metrics with subscript *c* refer to those computed from the *c*-th binary CM, obtained by reducing the multiclass CM fixing the *c*-th class. And metrics with subscript *w* denote weighted metrics, which consider the individual contributions of each class by weighting the metric value of the *c*-th class by the number of samples, $M_{c}$, of class *c*. This approach provides a “fair” evaluation by considering the impact of imbalanced class distributions on the overall performance.

Furthermore, in order to quantify how much compression is achieved via the LRMV-DTF layer, a compression ratio, Γ, between the number of parameters in the MV-DTF layer and those in the LRMV-DTF layer, i.e., *L* and $\hat{L}$, is defined in Equation (Equation 33). Note that Γ is independent of the latent space dimension, and it depends only on the view dimensions. It ensures that the compression ratio is consistent, regardless of the latent tensor space dimension.
(33)$$\Gamma =\frac{L}{\hat{L}}=\frac{J\times \left(\prod_{m=1}^{M}I_{m}+1\right)}{J\times \left(R\sum_{m=1}^{M}I_{m}+1\right)}=\frac{\prod_{m=1}^{M}I_{m}+1}{R\sum_{m=1}^{M}I_{m}+1}$$

### 6.4. Hyperparameter Tuning: The Latent Space Dimension *J* and the Rank *R* Values

To determine suitable hyperparameters for the low-rank MV-DTF layer, cross-validation via a grid search was employed [128]. Let R,J⊂N be two finite sets containing candidate values for the rank *R* and latent space dimensionality *J*, respectively. A grid search trains the multitask, multi-view model, built with the pair (J,R)∈J×R, on the training set $D_{tr}^{\left(t\right)}$, and it subsequently evaluates its performance on the validation set $D_{va}^{\left(t\right)}$ using some metric, M. The most suitable pair of values $\left(J^{*},R^{*}\right)$ is that which achieves the highest performance metric M over the validation set $D_{va}^{\left(t\right)}$.

For our case study with a tensor, $A\in R^{J\times 4\times 3\times 2}$, we fixed J={2,4,8,16,32,64,128,256} to study the impact of the *J* value on the classification metrics across tasks, whereas R={1,2} was selected based on the rank values that reduce the number of parameters in the LRMV-DTF layer according to the compression ratio (see Table 7), and R=3 and R=4 for performance analysis only. Since our datasets exhibit class imbalance, the MCC as the evaluation metric was used, given its robustness on imbalanced classes, as explained by Luque et al. in [127]. Through this empirical process, we found that J=16 and R=2 achieve the best trade-off between model performance and the compression ratio Γ in the set J×R. The sets J and R, and the most suitable pair of values, $(J^{*},R^{*})\in J\times R$, must be determined for each multitask, multi-view dataset.

### 6.5. Performance Evaluation

In this section, the performance of our multitask, multi-view case study in occlusion detection and vehicle classification tasks is evaluated. Our experiments focused on evaluating the impact of the rank, *R*, and dimension, *J*, of the latent tensor space S on computational complexity and model performance. To ensure the consistency of our results, each experiment was repeated 30 times. We first provide the results for the space saving achieved using different *R* and *J* values in MV-DFT and its low-rank approximation, LRMV-DTF, followed by an analysis of their effects on the learning phases and model performance.

Table 7 provides a comparison of the Γ compression achieved across different pairs of (J,R) values and two multi-view input space dimensions. It is noteworthy that compression is only achieved for Γ>1, and the larger the Γ, the higher the compression. Specifically, for the multi-view space $R^{4}\times R^{3}\times R^{2}$, compression is achieved only for R≤2 (see Figure 8a), while for the multi-view space $R^{40}\times R^{30}\times R^{20}$, a compression can be achieved for higher rank values (see Figure 8b). In consequence, Γ=1 provides an upper rank bound, denoted as $R_{max}$, beyond which compression is no longer achieved. For tensors with greater dimensions or a greater order, the upper rank bound would be greater (see Figure 8).

Figure 8 illustrates the relationship between the compression ratio Γ and the rank *R* for various multi-view spaces with different order and dimensionalities. For each space, we observe that, when the rank *R* increases, the compression ratio Γ decreases. Figure 8a,b show the compression Γ for multi-view spaces with the same order but different dimensionality, while Figure 8c,d show the compression Γ for higher-order multi-view spaces.

Figure 9 shows the training and validation loss curves over epochs for the model using either the MV-DTF or LRMV-DTF layer. From this figure, distinct behaviors in the loss curves can be observed on the training and validation phases:For low-dimensional latent tensor space (see Figure 9a,d), although stable, a slower convergence and higher loss values for both training and validation are observed. This indicates that the model may be underfitting.For high-dimensional latent tensor space (Figure 9c,f), a lower training loss is achieved. However, it exhibits fluctuations in the validation loss, especially for Task 2 (Figure 9f). This suggests that the model begins to overfit as *J* increases, leading to probable instability in validation performance. The marginal gains in training loss do not justify the increased risk of overfitting.For J=16 (Figure 9b,e), the most balanced performance across both tasks is achieved, showing faster convergence and smoother validation loss curves compared to J=2 and J=64. It achieves lower training loss while maintaining minimal validation loss variability, indicating good generalization.

In the subsequent subsections, the performance evaluation for each task on the tested videos is presented, highlighting the impact of the selected hyperparameters in the model’s generalization.

#### 6.5.1. Vehicle Occlusion Detection Results

This section presents the comparison results of the proposed multitask, multi-view model, with either the MV-DTF or LRMV-DTF layer and different pair of (J,R) values, on the occlusion detection task. Figure 10 shows the mean values of performance metrics obtained from our model for 30 different training runs, evaluated across test videos. Each row corresponds to a specific test video, while each column reflects a particular latent tensor-space dimension *J* value. As illustrated in Figure 10, a performance drop for different rank *R* values is very low, especially in high-dimensional spaces (e.g., J=16 and J=64, on the second and third columns of Figure 10). However, in low-dimensional spaces (see the first column of Figure 10 for J=2), the rank choice has a slightly greater impact on the performance, and a fine-tuning rank *R* value is necessary, as the dimension *J* decreases.

Additionally, Figure 10 is complemented by Table A1, which presents the mean and standard deviation of performance metrics across multiple runs, with the best and worst values highlighted in blue and red, respectively. From this table, the pair (16,2) shows the lowest standard deviations across most metrics, providing a good balance between computational complexity (see Table 7) and competitive performance with the MV-DTF layer. Although high-dimensional spaces (e.g., J=64) yield high performance, they also tend to exhibit large standard deviations, potentially increasing the risk of overfitting despite their higher mean values.

Finally, Figure 11 presents a performance comparison between our multi-view multitask model, using the LRMV-DTF with (16,2), and single-task learning (STL) single-view learning (SVL) models of SVM and RF, tested across test videos. Figure 11 highlights that the proposed model exhibits higher and more consistent performance than STL-SVL models on all metrics and videos, particularly in V2, V3, and V4. In contrast, the SVM and RF models show a noticeable performance drop in these videos. Overall, the proposed model improves the performance in the MCCnw metric of up to 92.81%, which represents a significant 6% improvement over the SVM and RF models.

#### 6.5.2. Vehicle-Size Classification Results

This section presents the comparison results of the proposed multitask, multi-view model, incorporating either the MV-DTF or LRMV-DTF layer with different pairs of (J,R) values, on the vehicle-size classification task. Figure 12 shows the mean values of the performance metrics for 30 different training runs on the test videos, where each row and column are related to a specific test video, and latent tensor space dimension, respectively. From this figure, we observe that the lower the *J* value, the worse the performance. Similarly, high *R* values generally contribute to improved performance. For J=2, there is a noticeable drop in performance, especially on the GMw, BMnw, and MCCnw metrics, suggesting that low-dimensional spaces fail to capture the complexity of the task. However, as long as *J* increases to 16 and 64, the metrics stabilize, and the performance drops across ranks becomes negligible, particularly for the ACCw and F1w metrics.

Additionally, Table A2 shows the mean and standard deviation of performance metrics across runs, with the best and worst values highlighted in blue and red, respectively. From this table, we found that high-dimensional spaces tend to yield not only higher mean performance but also lower standard deviation, indicating more stable and consistent outcomes across different test videos. In contrast, low-dimensional spaces (e.g., J=2) are more sensitive to the rank *R* hyperparameter, particularly for GMw, BMnw, and MCCnw. Consequently, a computationally efficient LRMV-DTF layer can be achieved in high-dimensional spaces by selecting low rank values without a significant performance drop. In contrast, for low-dimensional latent spaces (J=2), the performance is more sensitive to the choice of *R*, especially for GMw, BMnw, and MCCnw. Therefore, selecting an appropriate rank becomes crucial for low *J* values to avoid significant drops in performance.

Finally, in Figure 13, a comparison between our multi-view, multitask model and STL-SVL models (SVM and RF) across test videos is presented. This figure highlights the superiority of our multitask model, particularly in videos V3 and V4, where the SVM and RF models again exhibit a significant performance drop. Overall, the proposed model improves the performance in the MCCnw metric by up to 95.10%, which represents a significant 7% improvement over the SVM and RF models.

#### 6.5.3. Comparison with a Multitask Single-View Model

We also provide a comparison between the proposed multitask, multi-view model with its corresponding single-view model in Table A3. The latter model is basically the proposed model but without incorporating the MV-DTF layer, and the input space can only be either $X_{1}$, $X_{2}$, or $X_{3}$. However, in this work, we fix the input space to $X_{2}$. Finally, for a fair comparison, this model incorporates a layer that maps the feature space $X_{2}$ onto a latent space of dimension *J*.

The results provided in Table A3 show the overall mean value of weighted metrics across all videos, where it can be observed that incorporating the MV-DTF layer into this single-view model an improvement of up to 1.73% and 1.1% on the BMnw and MCCnw metrics, is achieved. These results are also consistent across all latent space dimensions.

Unlike the F1 metric, the experimental results show that the performance of single-view models does not exhibit a negative impact when the fusion layer is incorporated. Furthermore, even though the model parameters increase, incorporating the MV-DTF layer offers several advantages, including that the layer approximation through Hadamard products allows selecting ranks that, unlike the classical CPD, higher compression rates can be achieved.

Finally, Figure A1 and Figure A2 show the results for the occlusion detection and vehicle-size classification tasks, respectively. In contrast to the performance shown in Figure 10 and Figure 12, Figure A1 and Figure A2 show each metric independently for more detail.

### 6.6. Discussion

The promising results of the MV-DTF layer and its low-rank approximation LRMV-DTF comprise the following:The performance and consistency of the multitask, multi-view model are significantly influenced by the dimensionality of the latent tensor space (see Figure 10 and Figure 12). For a specific dimension, $J^{*}$, the model exhibits two distinct behaviors, given another *J* value: for $J\leq J^{*}$, the model tends to underfit, whereas for $J>J^{*}$, it is prone to overfitting to the training data.A negligible performance drop was observed in our case study as the compression ratio Γ approaches 1, i.e., $\hat{L}\to L$, when the LRMV-DTF layer is employed. This result provides empirical evidence of the underlying low-rank structure in the subtensors $A_{j:\cdot :}$ of tensor A in the MV-DTF layer.The maximum allowable rank value $R_{max}$ (upper rank bound) that achieves parameters’ compression increases as the dimensions of the multi-view space grow and/or as the number of dimensions (tensor order) increases.

The major limitations of the MV-DTF layer are as follows:Selecting suitable hyperparameters, i.e., the dimensionality $J_{1}\times \cdots \times J_{P}$ or *J* of the latent tensor space S, and the rank *R* for the LRMV-DTF layer, is a challenging task.A high-dimensional latent space increases the risk of overfitting, while very low-dimensional spaces may not fully capture the underlying relationships across views, resulting in underfitting.Reducing the rank of subtensors tends to decrease performance and increase the risk of underfitting classification models for low-dimensional latent spaces. Although higher rank values may improve model performance, they also increase the risk of overfitting.The choice of rank *R* involves a trade-off: higher values increase computational complexity but can capture more complex patterns, while lower values reduce the computational burden but may limit expressiveness of the model, resulting in performance decreasing.

## 7. Conclusions

In this work, we found a novel connection between the Einstein and Hadamard products for tensors. It is a mathematical relationship involving the Einstein product of the tensor $A\in R^{J\times I_{1}\times \cdots \times I_{M}}$ associated with a multilinear map $T:X_{1}\times \cdots \times X_{M}\to S$, and a rank-one tensor $X=x^{\left(1\right)}⊗\cdots ⊗x^{\left(M\right)}$, where dim(S)=J, $\dim\left(X_{m}\right)=I_{m}$ for all m∈[M], and $x^{\left(m\right)}\in X_{m}$. By enforcing low-rank constraints on the subtensors of A, which result by fixing every index but the last *M*, each *j*-th subtensor $A_{j:\cdots :}$ is approximated as a rank-$R^{\left(j\right)}$ tensor through the CPD. By exploiting this structure, a set of *M* third-order tensors $U^{\left(1\right)},\cdots ,U^{\left(M\right)}$, here called the Hadamard factor tensors, are obtained. We found that the Einstein product $A{⊛}_{M}X$ can then be approximated by a sum of *R* Hadamard products of *M* Einstein products $U_{:r:}^{\left(m\right)}{⊛}_{1}x^{\left(m\right)}$, where *R* corresponds to the maximum decomposition rank across subtensors, and $U^{\left(m\right)}\in R^{J\times R\times I_{m}}$ for all m∈[M].

Since multi-view learning leverages complementary information from multiple feature sets to enhance model performance, a tensor-based data fusion layer for neural networks, called Multi-View Data Tensor Fusion, is here employed. This layer projects *M* feature spaces $X_{1},\cdots ,X_{M}$, referred to as views, into a unified latent tensor space S through a mapping $g:X_{1}\times \cdots \times X_{M}\to S$, where dim(S)=J, and $\dim\left(X_{m}\right)=I_{m}$ for all m∈[M]. Here, we constrain *g* to the space of affine mappings composed of a multilinear map, $T:X_{1}\times \cdots \times X_{M}\to S$, followed by a translation. The multilinear map is here represented by the Einstein product $A{⊛}_{M}X$, where $A\in R^{J\times I_{1}\times \cdots \times I_{M}}$ is the induced tensor of T, and $X\in X_{1}⊗\cdots ⊗X_{M}$. Unfortunately, as the number of views increases, the number of parameters that determine *g* grow exponentially, and consequently, its computational complexity also grows.

To mitigate the curse of dimensionality in the MV-DTF layer, we exploit the mathematical relationship between the Einstein product and Hadamard product, which is the low-rank approximation of the Einstein product, useful when the compression ratio Γ>1.

The use of the LRMV-DTF layer based on the Hadamard product does not imply necessarily an improvement of the model performance compared to the MV-DTF layer based on the Einstein product. In fact, the dimension of the latent space *J* and the rank of subtensors *R* must be tuned via cross-validation (see Section 6.4). When the decomposition rank of subtensors is less than the upper rank bound $R_{max}$ (Γ>1), an efficient low-rank approximation of the MV-DTF layer based on the Einstein product is obtained.

From our experiments, we show that the intoduction of the MV-DTF and LRMV-DTF layers in a case study multitask VTS model for vehicle-size classification and occlusion detection tasks improves its performance compared to single-task and single-view models. For our case study, i.e., a particular case using the LRMV-DTF layer with J=16 and R=2, our model achieved an MCCnw of up to 95.10% for vehicle-size classification and 92.81% for occlusion detection, representing significant improvements of 7% and 6%, respectively, over single-task single-view models while reducing the number of parameters by a factor of 1.3.

Finally, for every case study, the dimension of the latent tensor space, *J*, and the decomposition rank, *R*, must be tuned. Additionally, the employment of an MV-DTF layer or a LRMV-DTF layer must be determined while the tradeoff between the model performance and computational complexity is taken into account.

### Open Issues

A computational complexity analysis must be conducted to evaluate the LRMV-DTF layer efficiency.For VTS systems, to integrate other high-dimensional feature spaces in order to improve the expressiveness of the latent tensor space and its computational efficiency.To explore other tensor decomposition models, such as the tensor-train model, for more efficient algorithms in high-dimensional data.To extend our work to more complex network architectures.To address other VTS tasks within the MTL framework for a more comprehensive vehicle traffic model.

## Figures and Tables

**Figure 1 sensors-24-07463-f001:**
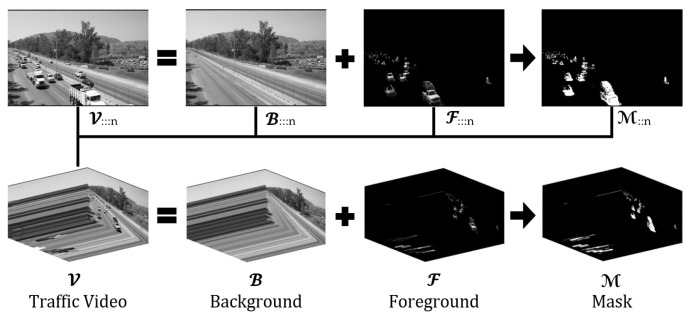
Illustration of the traffic surveillance video decomposition model.

**Figure 2 sensors-24-07463-f002:**
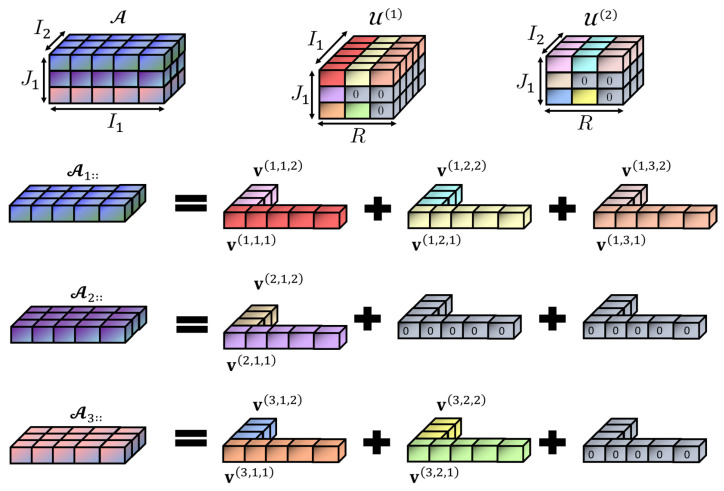
Illustration of subtensors of A and the Hadamard factor tensors $U^{\left(1\right)}\in R^{J\times R\times I_{1}}$, and $U^{\left(2\right)}\in R^{J\times R\times I_{2}}$ for the multilinear map $T:R^{I_{1}}\times R^{I_{2}}\to R^{J}$, where $I_{1}=5$, $I_{2}=3$, J=3, and R=3.

**Figure 3 sensors-24-07463-f003:**
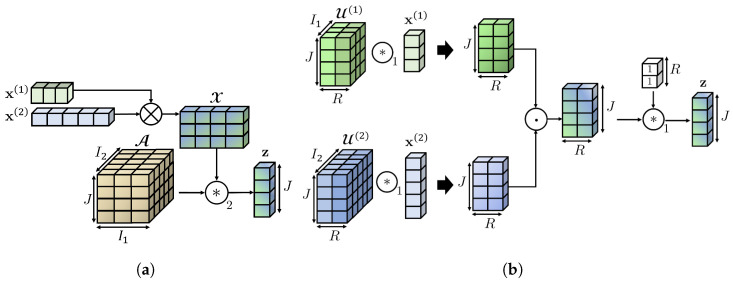
Illustration of the MV-DTF and LRMV-DTF layers. (**a**) MV-DTF layer $z=A{⊛}_{1}X$, where $X=x^{\left(1\right)}⊗x^{\left(2\right)}$. (**b**) LRMV-DTF layer $z=\left(\left(U^{\left(1\right)}{⊛}_{1}x^{\left(1\right)}\right)⊙\left(U^{\left(2\right)}{⊛}_{1}x^{\left(2\right)}\right)\right){⊛}_{1}1^{R}$.

**Figure 4 sensors-24-07463-f004:**
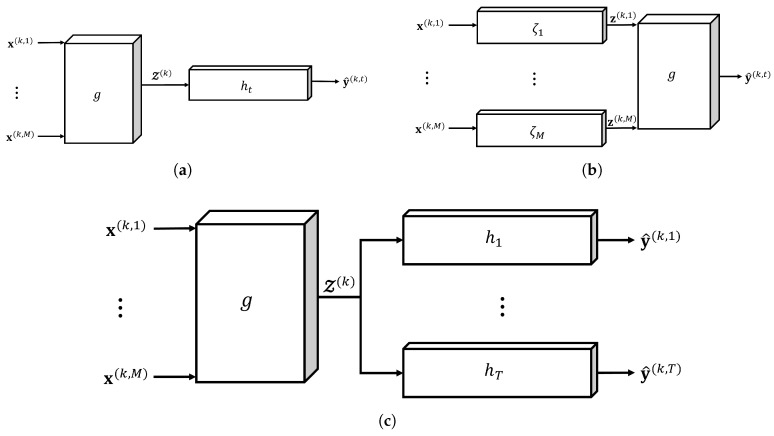
Primary configurations for incorporating the MV-DTF layer in neural network architectures. (**a**) MV-DTF layer for multi-view feature extraction on single-task learning, where $g:R^{I_{1}}\times \cdots \times R^{I_{M}}\to R^{J_{1}\times \cdots \times J_{P}}$, and $h:R^{J_{1}\times \cdots \times J_{P}}\to R^{O_{t}}$. (**b**) MV-DTF layer for multilinear regression on single-task learning, where $\zeta_{m}:R^{I_{m}}\to R^{H_{m}}$, and $g:R^{H_{1}}\times \cdots \times R^{H_{M}}\to R^{O_{t}}$. (**c**) MV-DTF for multi-view feature extraction on multitask learning, where $g:R^{I_{1}}\times \cdots \times R^{I_{M}}\to R^{J_{1}\times \cdots \times J_{P}}$, and $h_{t}:R^{J_{1}\times \cdots \times J_{P}}\to R^{O_{t}}$.

**Figure 5 sensors-24-07463-f005:**
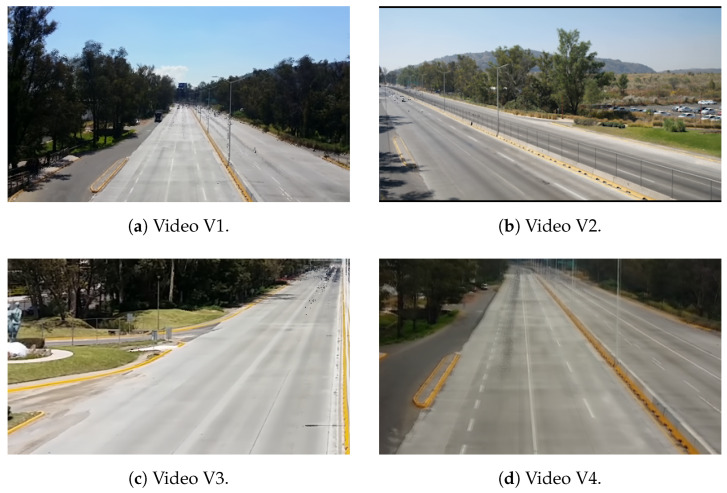
Test videos employed for vehicle-size classification and occlusion detection tasks. They were recorded at different dates and views (click on each Figure to link to the video).

**Figure 6 sensors-24-07463-f006:**
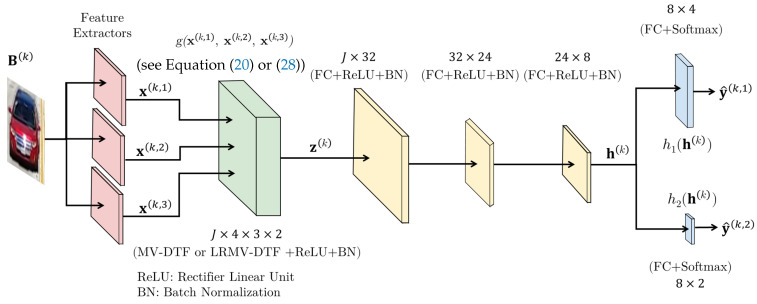
The proposed multitask, multi-view ANN architecture.

**Figure 7 sensors-24-07463-f007:**
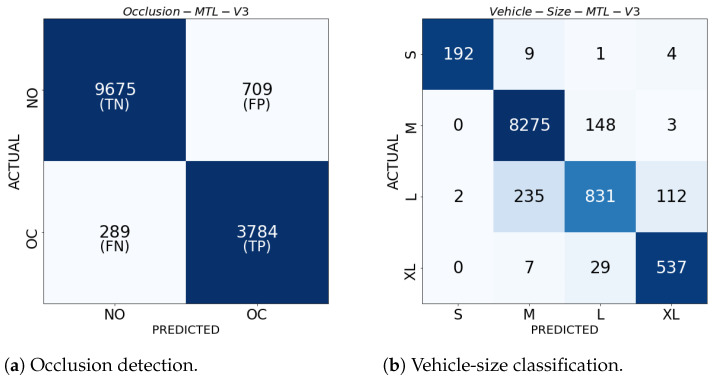
Confusion matrices for vehicle-size classification and occlusion detection on video V3 (J=2 and R=2), whose entries correspond to the mean values of runs.

**Figure 8 sensors-24-07463-f008:**
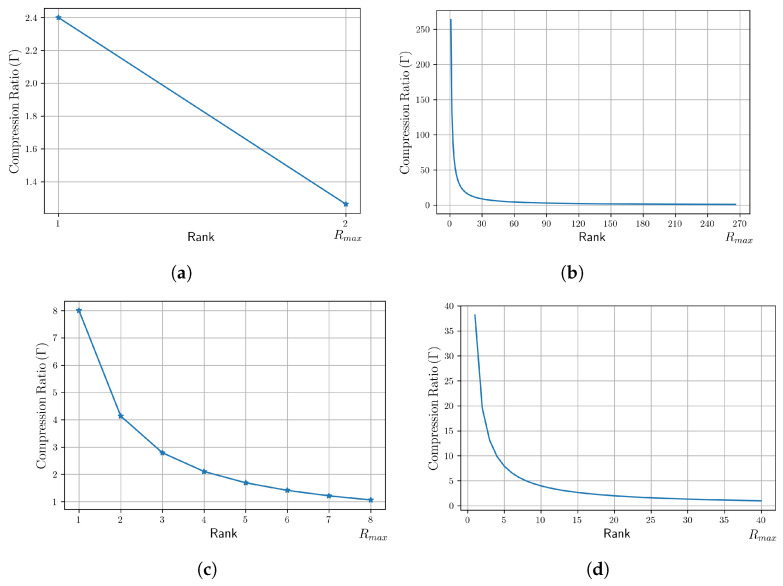
Compression ratio Γ for the set of rank *R* values that enable compression (Γ>1) and multi-view input space dimensions. (**a**) $\dim(X_{1}\times X_{2}\times X_{3})=4\times 3\times 2$ our case study. (**b**) $\dim(X_{1}\times X_{2}\times X_{3})=40\times 30\times 20$. (**c**) $\dim(X_{1}\times X_{2}\times X_{3}\times X_{4})=4\times 3\times 2\times 5$. (**d**) $\dim(X_{1}\times X_{2}\times X_{3}\times X_{4}\times X_{5})=4\times 3\times 2\times 5\times 7$.

**Figure 9 sensors-24-07463-f009:**
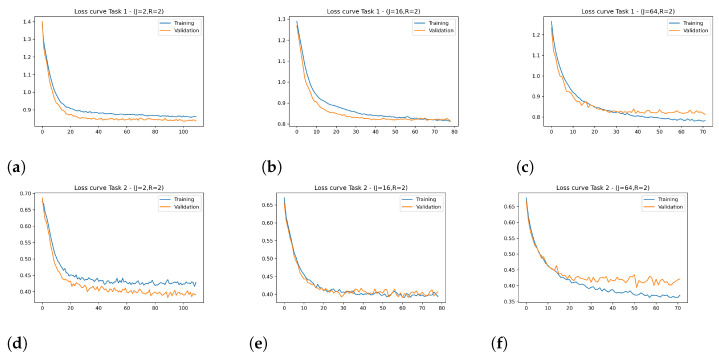
Loss curves obtained during the training and validation stages of the multitask, multi-view model across different latent space dimensions (J values), with fixed rank R = 2, for occlusion detection (first row) and vehicle-size classification (second row). (**a**) Loss curves for task 1 (J=2). (**b**) Loss curves for task 1 (J=16). (**c**) Loss curves for task 1 (J=64). (**d**) Loss curves for task 2 (J=2). (**e**) Loss curves for task 2 (J=16). (**f**) Loss curves for task 1 (J=64).

**Figure 10 sensors-24-07463-f010:**
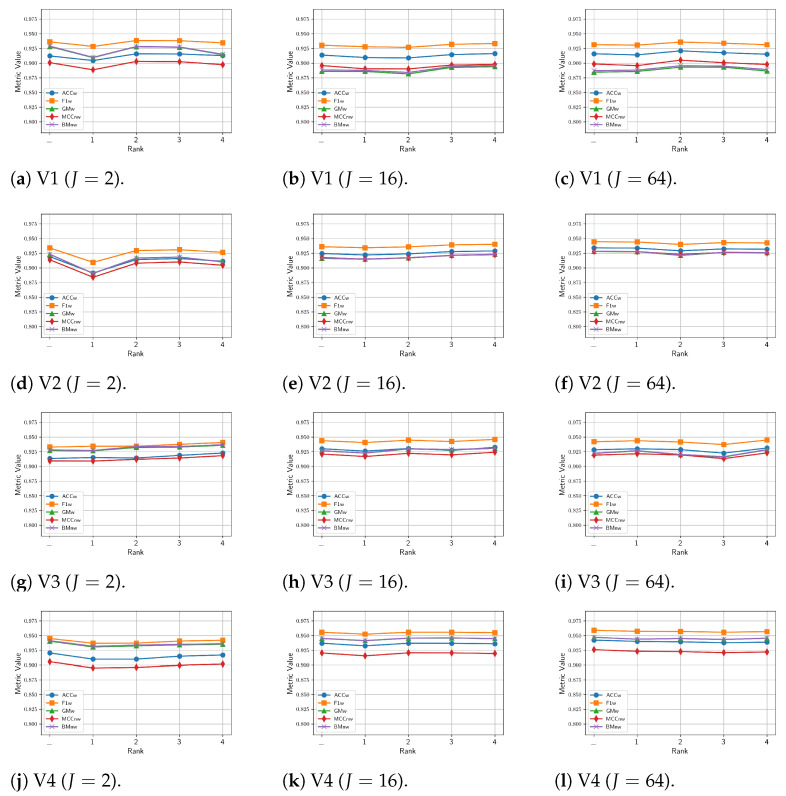
Mean values for 30 runs of performance metrics achieved on the occlusion detection task in test videos with the multi-view multitask model. The value R=− denotes the results when the MV-DTF layer is employed, whereas the other values correspond to the LRMV-DTF layer.

**Figure 11 sensors-24-07463-f011:**
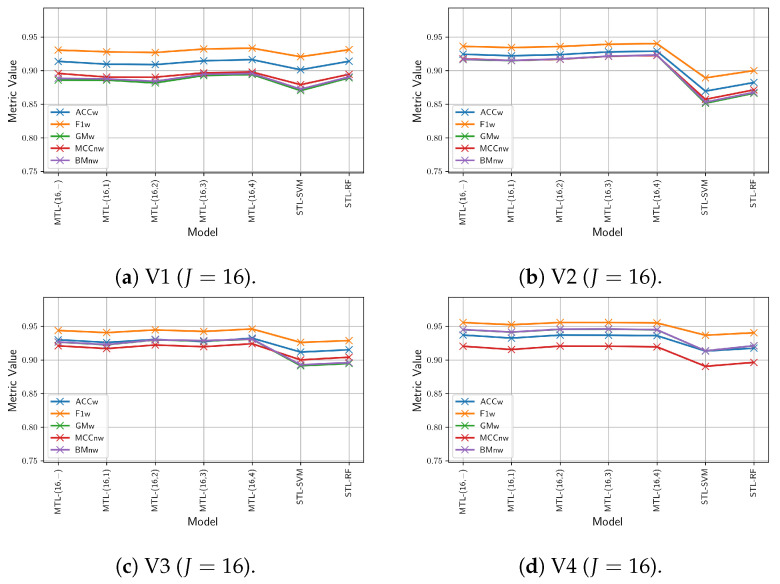
Comparison results between MTL and STL models on the occlusion detection task.

**Figure 12 sensors-24-07463-f012:**
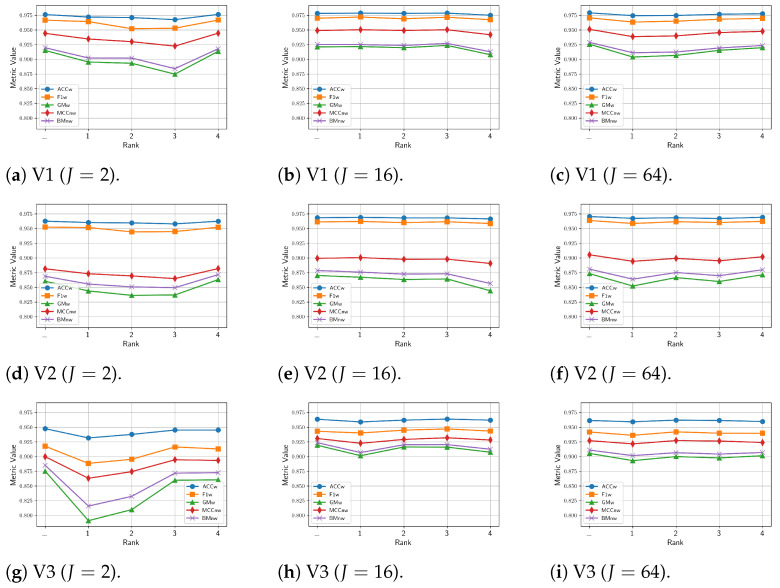
Mean values for 30 runs of performance metrics achieved on the vehicle-size classification task in test videos with the multi-view multitask model. The value R=− denotes the results when the MV-DTF layer is employed, whereas the other values correspond to the LRMV-DTF layer.

**Figure 13 sensors-24-07463-f013:**
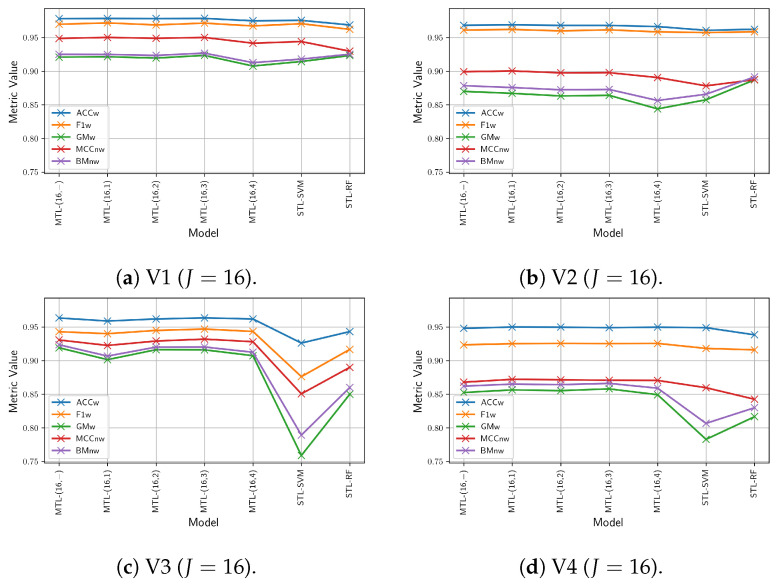
Comparison results between MTL and STL models on the vehicle-size classification task.

**Table 2 sensors-24-07463-t002:** Basic notation used in this work.

R,N,B	The field for real, natural, and binary numbers
≃	It denotes isomorphism between two structures
[N]	The subset of natural numbers {1,⋯,N}⊂N
X,X,x,x	Tensor, matrix, column vector, and scalar
dim(V)	The dimension of a vector space, *V*
⊙	Hadamard product
⊗	Tensor product
$\times_{n}$	The n-mode tensor-matrix product
${⊛}_{N}$	The Einstein product along the last *N* modes
$X^{\left(i_{1},\cdots ,i_{M}\right)}$	The $\left(i_{1},\cdots ,i_{M}\right)$th element of a sequence, $\left\{X^{\left(1,\cdots ,1\right)},\cdots ,X^{\left(I_{1},\cdots ,I_{M}\right)}\right\}$
	indexed by $i_{1}\in \left[I_{1}\right],\cdots ,i_{M}\in \left[I_{M}\right]$, where *X* can be a scalar, vector, or tensor
$X_{m}$	The feature space for the *m*-th view
$Y_{t}$	The output space for the *t*-th task
$f_{t}$	The *t*-th classification task
$g,\hat{g}$	The MV-DTF layer and its low-rank approximation
$h_{t}$	The *t*-th task-specific function
$H_{t}$	Hypothesis space of the classifiers for the *t*-th task
S	The latent tensor space
*P*	The order of the latent tensor space S
$J_{1}\times \cdots \times J_{P}$	The dimension of the latent tensor space S
$R^{\left(j_{1},\cdots ,j_{P}\right)}$	For a tensor $A\in R^{J_{1}\times \cdots \times J_{P}\times I_{1}\times \cdots \times I_{M}}$, it denotes the
	tensor rank of the $\left(j_{1},\cdots ,j_{P}\right)$-th subtensor $A_{j_{1}\cdots j_{P}:\cdots :}$

**Table 3 sensors-24-07463-t003:** Technical details of the test videos.

Video	Duration (s)	Tracked Vehicles	Temporal Samples of Tracked Vehicles
V1	146	137	6132
V2	326	333	19,194
V3	216	239	14,457
V4	677	720	91,870

**Table 4 sensors-24-07463-t004:** Description of the vehicle instance dataset for the occlusion detection task.

Video	Occluded	Unoccluded
V1	4671	1461
V2	12,684	6510
V3	10,384	4073
V4	41,002	11,084

**Table 5 sensors-24-07463-t005:** Description of the vehicle instance dataset for the vehicle-size classification task.

Video	Small (Class 1)	Midsize (Class 2)	Large (Class 3)	Very Large (Class 4)
V1	45	4018	390	218
V2	169	11,687	676	152
V3	206	8426	1179	573
V4	777	35,843	3101	1282

**Table 6 sensors-24-07463-t006:** Mathematical definition of classification performance metrics used in this work (metrics marked with * are biased by class imbalance [127]).

Metric	Equation	Weighted Metric
ACC *	$ACC_{c}=\frac{TP_{c}+TN_{c}}{TP_{c}+FN_{c}+TN_{c}+FP_{c}}$	$ACC_{w}=\frac{1}{M}\sum_{c=1}^{C}M_{c}ACC_{c}$
F1 *	$F1_{c}=2\cdot \frac{PRC_{c}\cdot SNS_{c}}{PRC_{c}+SNS_{c}}$	$F1_{w}=\frac{1}{M}\sum_{c=1}^{C}M_{c}F1_{c}$
MCC *	$MCC_{c}=\frac{TP_{c}\cdot TN_{c}-FP_{c}\cdot FN_{c}}{\sqrt{\left(TP_{c}+FP_{c}\right)\left(TP_{c}+FN_{c}\right)\left(TN_{c}+FP_{c}\right)\left(TN_{c}+FN_{c}\right)}}$	$MCC_{w}=\frac{1}{M}\sum_{c=1}^{C}M_{c}MCC_{c}$
GM	$GM_{c}=\sqrt{SNS_{c}\cdot SPC_{c}}$	$GM_{w}=\frac{1}{M}\sum_{c=1}^{C}M_{c}GM_{c}$
BM	$BM_{c}=SNS_{c}+SPC_{c}-1$	$BM_{w}=\frac{1}{M}\sum_{c=1}^{C}M_{c}BM_{c}$
SNS	$SNS_{c}=\frac{TP_{c}}{TP_{c}+FN_{c}}$	$SNS_{w}=\frac{1}{M}\sum_{c=1}^{C}M_{c}SNS_{c}$
SPC	$SPC_{c}=\frac{TN_{c}}{TN_{c}+FP_{c}}$	$SPC_{w}=\frac{1}{M}\sum_{c=1}^{C}M_{c}SPC_{c}$
PRC *	$PRC_{c}=\frac{TP_{c}}{TP_{c}+FP_{c}}$	$PRC_{w}=\frac{1}{M}\sum_{c=1}^{C}M_{c}PRC_{c}$
Global GM	$GGM=\sqrt{SNS_{w}\cdot SPC_{w}}$	-
Global BM	$GBM=SNS_{w}+SPC_{w}-1$	-
multiclass MCC *	$mMCC=\frac{M\times \sum_{c=1}^{C}TP_{c}-\sum_{c=1}^{C}t_{c}p_{c}}{\sqrt{M^{2}-\sum_{c=1}^{C}p_{c}^{2}}\times \sqrt{M^{2}-\sum_{c=1}^{C}t_{c}^{2}}}$	-
	$t_{c}=\sum_{c=1}^{C}{\mathrm{CM}}_{c:}$	
	$p_{c}=\sum_{c=1}^{C}{\mathrm{CM}}_{:c}$	

**Table 7 sensors-24-07463-t007:** Compression ratio Γ for different pairs of (J,R) values and two multi-view spaces.

(J,R)	Input Space Dimension $\dim(X_{1}\times X_{2}\times X_{3})$					
	4×3×2 **(Our Case Study)**			40×30×20		
	L	$\hat{L}$	Γ	L	$\hat{L}$	Γ
(2, 1)	48	20	2.4	48,000	182	263.7
(2, 2)	48	38	1.26	48,000	362	132.6
(2, 3)	48	56	0.86	48,000	542	88.56
(2, 4)	48	74	0.65	48,000	722	66.48
(8, 1)	192	80	2.4	192,000	728	263.7
(8, 2)	192	152	1.26	192,000	1448	132.6
(8, 3)	192	224	0.86	192,000	2168	88.56
(8, 4)	192	296	0.65	192,000	2888	66.48
(32, 1)	768	320	2.4	768,000	2912	263.7
(32, 2)	768	608	1.26	768,000	5792	132.6
(32, 3)	768	896	0.86	768,000	8766	88.56
(32, 4)	768	1184	0.65	768,000	11,552	66.48

## Data Availability

The datasets used to support the findings of this study, particularly for the multi-view multitask model, are publicly available at the following GitHub repository: https://github.com/fhermosillo/VTSMultiviewDatasets (accessed on 20 October 2024). Unfortunately, the code employed in this work is currently private, but it can be made available upon request to reviewers for evaluation purposes. Please refer to the repository or contact the corresponding author for further inquiries or additional data requests.

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
