# Peer review of "A Tensor Space for Multi-View and Multitask Learning Based on Einstein and Hadamard Products: A Case Study on Vehicle Traffic Surveillance Systems"

_sensors, 2024, doi:10.3390/s24237463_

Round 1
Reviewer 1 Report
Comments and Suggestions for Authors
I carefully read this paper and did not find any obvious errors.
The theoretical results of this paper seem to be correct.
However, some details need attention. For examples,
1. Page 7 Equality (3): ``i_1'' should be ``i_1=1''.
2. Page 8 Equality (5): ``as Equation 5 shows.'' should be ``as Equation 5 shows:'', there should be a comma after (5), there should be no space before ``where ...''.
There are numerous similar questions in this manuscript, please revise them carefully.
Author Response
Please see in the attachment.

Reviewer 2 Report
Comments and Suggestions for Authors
In this work, the authors introduce a tensor-based method that can incorporate multi-view data and apply it to vehicle traffic surveillance image/video processing. They present two results: the first is their discovery of a relationship between an Einstein product and a Hadamard product and how this property can reduce the computational complexity of tensor calculations, and the other is their model of tensor-based multitask data fusion on video processing and traffic surveillance tasks. While the discovered mathematical property is valid, how this improves their proposed multitask model in video processing is not well demonstrated.
Major issues:
1. This manuscript is long due to the extensive introduction to tensor algebra and the theoretical concepts in vehicle traffic surveillance. While this may be appropriate for a textbook or a review article, for a work presenting primary discovery, introduced concepts should be used in the application. For example, concepts in Section 4 are common in video processing, not specific to their model nor helping explain what they actually do in Section 6.
2. While the proposed tensor low-rank approximation reduces the number of parameters to train, the prediction model depends on an ANN with at least 1000 parameters, which is a lot. In comparison, each input sample has only 9 features. It is hard to avoid overfitting with this model.
3. They stated that the primary result of this work is the relationship between Einstein and Hadamard products and how this property can help compress trainable parameters. However, their actual result shows that the said benefit only happens when the rank is 1 or 2.
4. The model shown in Fig. 6 should be described in more detail. This at least includes how the feature extractors work, how the training was performed with two separate outputs, and what specific loss function was used.
5. They benchmarked their model against SVM and RF. Would it be possible for the performance advantage to come from ANN, not the tensor data fusion?
6. The work should better explain their process of splitting training/validation/test sets regarding the temporal dependency of video-formatted input data. For example, can there be a case where a training image and a test image are on temporally adjacent frames? Given that they can be very similar, there may be a risk of data leakage.
Minor issues:
1. The figure and table captions in the manuscript should contain enough introduction and be self-sufficient to deliver the main point. For example, in Tbl. 7, it would be helpful in the table caption to annotate what L and Γ are even when you have defined them in the text.
2. Some of the figures are not well annotated. For example, the axes in Fig. 8 have no labels.
3. Figs. 9 and 10, it does not make sense to plot different metrics within one plot. They should be reorganized so different J’s can be compared directly within a plot.
4. Tbls. 8 and 9, having an entire page of numbers is not an effective way of delivering the message. Consider showing them in the plot or moving them to supplement.
5. Figures should be numbered in the order in which they appear in the text.
6. Typos and confusion in the math. Just to list a few here, not exhaustive:
a. Line 224, dim(V_m), not dim(V_M)
b. Line 261, RHS should be B_1 and B_2, not X_1 and X_2
c. Eq. (9), there should not be task-specific x, i.e. x^(k,m), not x^(k,m,t)
d. Sec. 4.1, V was set to have the B dimension, but it is not used later
Author Response
Please see in the attachment

Reviewer 3 Report
Comments and Suggestions for Authors
please see the attachment

please see the attachment
Author Response
Please see in the attachment.

Round 2
Reviewer 2 Report
Comments and Suggestions for Authors
After revision, the description of the method, especially in Sections 6.2.1 and 6.2.2, has been improved. Some presentations of results can still be refined in Figs. 9 and 10 to better show the results, as mentioned in my comment in the last review.
My main concern with this work is on the claim that the tensor product property will improve its application on the multi-view multitask model, as it requires using an ANN that contains many more parameters than what the tensor method improvement saves (even with dropouts). If they can demonstrate the performance of this model with a dataset with more views, this would actually make their point that the improvement of the tensor method really helps.
Author Response
Comment 1: After revision, the description of the method, especially in Sections 6.2.1 and 6.2.2, has been improved. Some presentations of results can still be refined in Figs. 9 and 10 to better show the results, as mentioned in my comment in the last review.
Response 1: Thank you for pointing this out. We agree with this comment. Therefore, a paragraph was added in Page 29, Line 869 "Finally, Figures A1 and A2 show the results for the occlusion detection and vehicle-size classification tasks, respectively. In contrast to the performance shown in Figures 10 and 12, Figures A1 and A2 show each metric independently for more detail.". Figures A1 and A2 have been added in Page 35 Line 1046 in order to complemet the answer to the suggestion.
Comment 2: My main concern with this work is on the claim that the tensor product property will improve its application on the multi-view multitask model, as it requires using an ANN that contains many more parameters than what the tensor method improvement saves (even with dropouts). If they can demonstrate the performance of this model with a dataset with more views, this would actually make their point that the improvement of the tensor method really helps.
Response 2: Thank you for pointing this out. We agree with this comment. Therefore, Section 6.5, Page 24, Table 7 between lines 771 and 772 has been extended; Figure 8 between lines 771 and 772 has been added.
Section 6.5, page 24, has been improved in lines 764-771 "Table 7 provides a comparison of the Γ compression achieved across different pairs of (J, R) values, and two multi-view input space dimensions. It is noteworthy that compres- sion is only achieved for Γ > 1, and the larger the Γ the higher the compression. Specifically, for the multi-view space R4 × R3 × R2, a compression is achieved only for R ≤ 2 (see Figure 8a), while for the multi-view space R40 × R30 × R20, a compression can be achieved for higher rank values (see Figure 8b). In consequence, Γ = 1 provides an upper rank bound, denoted by Rmax, beyond which compression is no longer achieved. For tensors with greater dimensions or order, the upper rank bound would be greater (see Figure 8)."
Section 6.5, page 24, line 772-777, a paragraph was added: "Figure 8 illustrates the relationship between the compression ratio Γ and the rank R for various multi-view spaces with different order and dimensionalities. For each space, we observe that when the rank R increases, the compression ratio Γ decreases. Figures 8a and 8b show the compression Γ for multi-view spaces with the same order with but different dimensionality, while Figures 8c and 8d show the compression Γ for higher-order multi-view spaces."
Conclusions, Page 30, Line 924-932, 936, Some conclusions have been improved for clarity between the MV-DTF layer. "To mitigate the curse of dimensionality in the MV-DTF layer, we exploit the mathe- matical relationship between the Einstein product and Hadamard product, which is the
low-rank approximation of the Einstein product, useful when the compression ratio Γ > 1.
The use of the LRMV-DTF layer based on the Hadamard product does not imply
necesarily an improvement of the model performance compared to the MV-DTF layer
based on the Einstein product. In fact, the dimension of the latent space J and the rank of
subtensors R must be tuned via cross-validation (see Section 6.4). When the decomposition
rank of subtensors is less than the upper rank bound Rmax (Γ > 1), an efficient low-rank
approximation of the MV-DTF layer based on the Einstein product is obtained.";
"For our case study, i.e., a particular case,"
Minor changes, Section 6.4, Page 23, lines 744-745, 749-750, 753-754, has been improved for clarity "To determine suitable hyperparameters for the low-rankMV-DTF layer, cross-validation via grid search is employed [131]. Let R, J ⊂ N be two finite sets containing candidate values for the rank R and latent space dimensionality J, respectively. Grid search train the multitask multi-view model, built with the pair (J, R) ∈ J × R, on the training set D(t) tr , and subsequently evaluates its performance on the validation set D(t) va using some metric M. The most suitable pair of values (J∗, R∗) is that who achieves the highest performance metric M over the validation set D(t) va ."
"For our case study, a tensor A ∈ RJ×4×3×2, we fix J = {2, 4, 8, 16, 32, 64, 128, 256} to study the impact of the J value on the classification metrics across tasks, whereasR = {1, 2} was selected based on the rank values that reduce the number of parameters in the LRMV- DTF layer according to the compression ratio (see Table 7), and R = 3 and R = 4 for performance analysis only. Since our datasets exhibit class imbalance, the MCC as the evaluation metric was used, given its robustness on imbalanced classes as explained by Luque, et. al. in [129]. Through this empirical process, we found that J = 16 and R = 2 achieve the best trade-off between model performance and the compression ratio Γ in the set J ×R. The sets J and R, and the most suitable pair of values (J∗, R∗) ∈ J ×R must be determined for each multitask multi-view dataset."
